# Unleashing the Power of Pre-trained Language Models for Offline Reinforcement Learning

**Ruizhe Shi**[1][*]   **Yuyao Liu**[1][*]   **Yanjie Ze**[2]   **Simon S. Du**[3]   **Huazhe Xu**[124]

[1]IIIS, Tsinghua University   [2]Shanghai Qi Zhi Institute   [3]University of Washington   [4]Shanghai AI Lab

## Abstract

Offline reinforcement learning (RL) aims to find a near-optimal policy using pre-collected datasets. In real-world scenarios, data collection could be costly and risky; therefore, offline RL becomes particularly challenging when the in-domain data is limited. Given recent advances in Large Language Models (LLMs) and their few-shot learning prowess, this paper introduces **La**nguage Models for **Mo**tion Control (**LaMo**), a general framework based on Decision Transformers to effectively use pre-trained Language Models (LMs) for offline RL. Our framework highlights four crucial components: (1) Initializing Decision Transformers with sequentially pre-trained LMs, (2) employing the LoRA fine-tuning method, in contrast to full-weight fine-tuning, to combine the pre-trained knowledge from LMs and in-domain knowledge effectively, (3) using the non-linear MLP transformation instead of linear projections, to generate embeddings, and (4) integrating an auxiliary language prediction loss during fine-tuning to stabilize the LMs and retain their original abilities on languages. Empirical results indicate **LaMo** achieves state-of-the-art performance in sparse-reward tasks and closes the gap between value-based offline RL methods and decision transformers in dense-reward tasks. In particular, our method demonstrates superior performance in scenarios with limited data samples. Our project website is **lamo2023.github.io**.

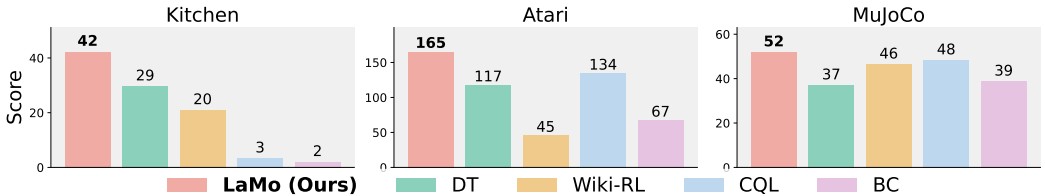

Figure 1: **Normalized score on D4RL (Fu et al., 2020) dataset** of **La**nguage Models for **Mo**tion Control (**LaMo**), Decision Transformer (DT, Chen et al., 2021), Wiki-RL (Reid et al., 2022), Conservative Q-Learning (CQL, Kumar et al., 2020) and Behavior Cloning (BC). We average scores over tasks and data sample ratios for each domain. (`Medium` for *Mujoco* and *Atari*, `Complete` and `Partial` for *Kitchen*, of different sample ratios, described in Appendix C.)

## 1 Introduction

Offline reinforcement learning (RL) has gained significant attention in recent years due to its potential for utilizing pre-collected datasets to improve agent performance (Lange et al., 2012; Prudencio et al., 2023; Levine et al., 2020). Among the prominent algorithms in offline RL, Decision Transformer (DT) (Chen et al., 2021) reframes RL as a conditional sequence modeling problem and utilizes the Transformer architecture (Vaswani et al., 2017), showing the potential of sequence models for decision making (Xu et al., 2022; Hu et al., 2023a;b; Xie et al., 2023; Laskin et al., 2023). However, Transformers are known to be data hungry (Khan et al., 2022; Brown et al., 2020; OpenAI, 2023), meaning that pre-training on massive amounts of data is usually required to achieve satisfactory model abilty (Touvron et al., 2021). One of the most pronounced applications of Transformers

---
[*]Equal contribution. Order is decided by coin flip.

— large language models (LLMs) — has achieved significant progress in language understanding recently, such as GPT (Radford & Narasimhan, 2018; Radford et al., 2019; Brown et al., 2020; OpenAI, 2023), ChatGPT (Ouyang et al., 2022), and LLaMA (Touvron et al., 2023a). Pre-trained on rich and diverse linguistic data, LLMs gain great few-shot and zero-shot learning abilities (Brown et al., 2020; Kojima et al., 2022).

A natural thought to enhance the Transformer-based sequential decision-making methods is thus to introduce the power of pre-trained Language Models (LMs) into them, initially explored by a lot of recent works (Ichter et al., 2022; Huang et al., 2022; Driess et al., 2023; Wu et al., 2023; Li et al., 2022; Reed et al., 2022; Lin et al., 2023; Brohan et al., 2023b;a; Tang et al., 2023; Wang et al., 2023b). Among them, Li et al. (2022) propose to encode the environment states with LLMs and learn a policy based on the decoded states, while their environment states are restricted to language descriptions only, making it hard for motion control. Reid et al. (2022) address this weakness by directly utilizing a pre-trained LM as the initialization of DT and processing low-level agent states and actions directly, instead of processing language descriptions. Their architecture thus successfully utilizes pre-trained LMs in motion control tasks like locomotion (Fu et al., 2020). However, despite the novelty of the proposed method in Reid et al. (2022), they still do not fully unleash the power of LMs: their empirical performance is on par with pure DT methods and lags behind CQL (Kumar et al., 2020). We thus ask,

**Can we unleash the power of pre-trained LMs to solve sequential decision-making problems?**

In this work, we propose **La**nguage Models for **Mo**tion Control (**LaMo**), a framework to effectively utilize pre-trained LMs for offline RL. While the motivation is straightforward, it takes four crucial designs to empower LaMo: 1) pre-trained language model is used as the initial weight of DT; 2) the pre-trained weights are frozen and the model is fine-tuned with parameter-efficient finetuning method LoRA (Hu et al., 2022) on **0.7%** of the parameters; 3) we replace the input embeddings and the output linear projections with Multi-Layer Perceptrons (MLPs); 4) a language prediction loss function as an auxiliary objective. Consequently, we find that the four components combined can help LaMo preserve the prior knowledge and generalization ability acquired from the pre-training while adapting efficiently to the new domain of offline RL.

We conduct comprehensive experiments across three distinct environments: *Kitchen* (Gupta et al., 2019), *MuJoCo* (Todorov et al., 2012), and *Atari* (Bellemare et al., 2013), spanning 8 tasks altogether. These tasks range from sparse-reward to dense-reward, and from state inputs and image inputs. For each task, we evaluate performance under varying data ratios to examine the influence of sample amount on the outcomes. We observe that as is shown in Figure 1, LaMo surpasses both DT and value-based baselines in **sparse-reward** tasks; and in **dense-reward** tasks, our method significantly outperforms DT and closes the gap between value-based methods and DT-based methods. Especially, we find that when the data scale is limited (*e.g.*, 1% of the whole dataset), LaMo demonstrates much more powerful learning ability, which could be credited to inductive bias within pre-trained LMs.

Our contributions are three-fold:

- We propose LaMo, a novel offline RL framework that unleashes the power of pre-trained language models.
- To better utilize the cross-domain knowledge from language modeling, we propose 3 additional techniques including LoRA finetuning, non-linear MLP projections, and an auxiliary language loss. Each module is shown to contribute positively to the final results of LaMo.
- Through extensive experiments in 8 tasks across diverse domains, dataset scales, and reward densities, we demonstrate the superiority of LaMo over DT-based and value-based offline RL algorithms. Specifically, we find that LaMo could successfully handle the challenging low-data regime while DT could not. This highlights the great potential of our cross-domain pre-training for sequential modeling.

## 2 RELATED WORK

**Transformers for decision making.** Transformers have dominated the language tasks in the NLP community (Radford & Narasimhan, 2018; Radford et al., 2019; Brown et al., 2020; Devlin et al.,

2019) and also started to show potential in other domains, such as decision making. As one initial trial to introduce Transformers into reinforcement learning (RL), Decision Transformer (DT) (Chen et al., 2021) models the elements such as states and actions into a sequence, thus framing the RL problem into a sequence prediction problem. There are a lot of following works make improvements under the framework of DT (Xu et al., 2022; Hu et al., 2023b; Xie et al., 2023; Yamagata et al., 2023; Liu & Abbeel, 2023). For example, Prompt DT (Xu et al., 2022) appends demonstrations into the sequence to achieve generalization in new tasks; Xie et al. (2023) pre-train DT by leveraging future trajectory information; Q-learning DT (Yamagata et al., 2023) refines the return-to-go in training data using Q-values, thereby imbuing DT with Q-learning's proficiency in handling sub-optimal data. Agentic Transformer (Liu & Abbeel, 2023) addresses the issues of sub-optimality by using chain of hindsight to relabel the target returns, which achieves competitive performance compared with value-based methods. Trajectory Transformer (Janner et al., 2021) trains on sequences of discretized states, actions, and rewards, indicating a more direct solution. Our work focuses on utilizing the cross-domain knowledge, *i.e.* language pre-training, as privileged information to enhance DT-based methods, which thus is orthogonal to these works.

**Large Language Models** (LLMs) have been the most pronounced application of the Transformer architecture in recent years (Radford & Narasimhan, 2018; Radford et al., 2019; Brown et al., 2020; OpenAI, 2023; Devlin et al., 2019; Touvron et al., 2023a;b). Pre-trained on massive amounts of corpus, LLMs have shown surprising few-shot and even zero-shot ability in language tasks, such as GPT series (Radford & Narasimhan, 2018; Radford et al., 2019; Brown et al., 2020; OpenAI, 2023). To personalize LLMs for different downstream user applications with computational efficiency, researchers commonly utilize parameter-efficient finetuning techniques (Hu et al., 2022; Zhang et al., 2023a; Li & Liang, 2021; Lester et al., 2021; Liu et al., 2022; Wang et al., 2023a) to finetune LLMs. In this work, we use the GPT-2 architecture (Radford et al., 2019) as the backbone due to its affordability and use LoRA (Hu et al., 2022) for downstream finetuning.

**LMs for decision making.** The great success of LMs in language tasks also motivates researchers to explore the potential of LMs for decision making problems (Ichter et al., 2022; Huang et al., 2022; Driess et al., 2023; Wu et al., 2023). One line of works (Ichter et al., 2022; Huang et al., 2022; Driess et al., 2023; Wu et al., 2023) utilizes LMs for high-level task decomposition and task planning, while their low-level execution policy is learned or designed separately. Another line of works (Li et al., 2022; Reed et al., 2022; Lin et al., 2023; Brohan et al., 2023a; Tang et al., 2023; Wang et al., 2023b) exploits the representation and generalization power of pre-trained LMs. Li et al. (2022) adapt pre-trained LMs to generate policies for tasks where the inputs could be converted into word sequences and point out the significance of sequential structure of inputs; Lin et al. (2023) use a geometric feasibility planner to encourage LM to generate both mid-level and low-level plans given language instruction; and Tang et al. (2023) design prompts for LMs to encode language instructions. When multi-modal inputs are involved, one solution is transforming them into one common embedding space (Brohan et al., 2023a; Reed et al., 2022). For example, RT-2 (Brohan et al., 2023a) utilizes a Vision-Language Model pre-trained on massive language and vision-language data, and also represents actions as text tokens on the Robot-Action Fine-tuning stage; GATO (Reed et al., 2022) utilizes a Vision Transformer to encode the image inputs, and learns from a large multi-modal, multi-task dataset to perform various tasks all in one model.

The most relevant work to us is Wiki-RL (Reid et al., 2022), which also uses a pre-trained language model as the initialization of DT for offline RL. However, their empirical results are shown to be only close to DT and could not surpass CQL (Kumar et al., 2020). Therefore, our work tries to better unleash the power of pre-trained LMs for offline RL.

# 3 PRELIMINARIES

## 3.1 OFFLINE REINFORCEMENT LEARNING

We formulate reinforcement learning (RL) as a standard Markov Decision Process (MDP) with a tuple $(\mathcal{S}, \mathcal{A}, T, d_0, \mathcal{R}, \gamma)$, where $\mathcal{S}$ is the set of states $s \in \mathcal{S}$, $\mathcal{A}$ is the set of actions $a \in \mathcal{A}$, $\mathcal{T}$ is the transition distribution of form $T(s_{t+1}|s_t, a_t)$, $d_0(s_0)$ describes the distribution of states $s_0$, $\mathcal{R} : \mathcal{S} \times \mathcal{A} \to \mathbb{R}$ is the reward function, $r_t = \mathcal{R}(s_t, a_t)$ is the reward at timestep $t$, and $\gamma \in (0, 1)$ is the discount factor. The agent in this MDP follows a policy $\pi(a|s)$, and the objective is:

$$J(\pi) = \mathbb{E}_{s_0 \sim d_0(\cdot),\ a_t \sim \pi(\cdot|s_t),\ s_{t+1} \sim T(\cdot|s_t, a_t)} \left[ \sum_{t=0}^{\infty} \gamma^t \mathcal{R}(s_t, a_t) \right]. \tag{1}$$

In offline RL, the access to interacting with the environment is removed while the objective remains $J(\pi)$. Agents could only learn on pre-collected trajectories $\mathcal{D} = \{(s_t^{(i)}, a_t^{(i)}, s_{t+1}^{(i)}, r_t^{(i)})\}$, which is generated by a unknown behavior policy $\pi_B$. Here we introduce common properties of the dataset $\mathcal{D}$: *1)* **Sub-optimality.** In many contexts, $\pi_B$ is not an optimal policy, i.e., $\mathcal{D}$ would not contain the optimal behaviors, and thus simple imitation may exhibit suboptimal performance; *2)* **Dense-reward or sparse-reward.** In the dense-reward environment, agents receive reward signals that correspond to whether agents' behaviors are good for each timestep, while in the sparse-reward setting, positive reward signals from the environments might be only given when success is achieved, and otherwise are zero. The sparse-reward setting is thus much more challenging but closer to the real world scenarios.

## 3.2 DECISION TRANSFORMER

Following Decision Transformer (DT), we frame the RL problem as a sequential modeling problem. We consider each trajectory $\tau$ as a sequence of ordered return-to-go $\hat{R}$, action $a$, and states $s$, defined as follows,

$$\tau = \left(\hat{R}_{t_0}, s_{t_0}, a_{t_0}, \hat{R}_{t_0+1}, s_{t_0+1}, a_{t_0+1}, \ldots, \hat{R}_{t_0+K-1}, s_{t_0+K-1}, a_{t_0+K-1}\right). \tag{2}$$

where return-to-go $\hat{R}$ is defined as the sum of rewards from the current timestep to the future: $\hat{R}_k = \sum_{i=k+1}^{T} r_i$, $T$ is the episode length, and $K$ is the context length. The learning objective of the model is to predict the future action $a_t'$ given the history sequence and the current state $s_t$, while the ground truth is $a_t$, written as a simple squared error term:

$$\mathcal{L}_{\text{decision}} = \sum_{t=t_0}^{t_0+K-1} \|a_t - a_t'\|_2^2. \tag{3}$$

## 4 METHOD

We propose **La**nguage Models for **Mo**tion Control (**LaMo**), an effective framework that incorporates pre-trained Language Models (LMs) into offline Reinforcement Learning, to leverage the reasoning and few-shot ability of LMs and solve challenging scenarios such as limited data and sparse reward. An illustration of LaMo is given in Figure 2. LaMo encompasses several crucial designs: *1)* We adopt a pre-trained LM (*i.e.,* GPT-2 (Radford et al., 2019)) as the initialization of a Decision Transformer (DT) (Chen et al., 2021); *2)* We replace the linear embedding projections with MLPs to augment representation learning capabilities for complicated tasks; *3)* During training the offline RL agents, we freeze the pre-trained parts and utilize the parameter-efficient fine-tuning technique LoRA (Hu et al., 2022), where the trainable parameters account for only **0.7%** of the entire model; *4)* We introduce language prediction as an auxiliary objective while finetuning, in order to stabilize the performance and maintain the language ability.

Figure 2: **The overview of LaMo.** LaMo mainly consists of two stages: *1)* pre-training LMs on language tasks, *2)* freezing the pre-trained attention layers, replacing linear projections with MLPs, and using LoRA to adapt to RL tasks. We also apply the language loss during the offline RL stage as a regularizer.

### 4.1 PRE-TRAINING ON LANGUAGE TASKS

The initial step involves obtaining pre-trained language models (LMs). Considering the widespread recognition and computational affordability of the GPT-2 architecture (Radford et al., 2019), we

utilize the commonly available pre-trained weight of GPT-2 from Hugging Face[1]. To further explore the effects of the quality of different pre-trained models on the downstream offline RL tasks, we also pre-train GPT-2 by ourselves in the ablation study, using the corpus dataset WikiText (Merity et al., 2017) and the common next-token prediction objective

$$\mathcal{L}_{\text{language}} = \sum_{i=1}^{s-1} -\log\left(T\left(w_{i+1}|w_1,\ldots,w_i\right)\right), \tag{4}$$

where $w_i$ is the $i$th language token in one sentence, and $T$ is the probability distribution of next token predicted by the model. We have explored three variants of models: *1)* a model that is pre-trained for fewer steps; *2)* a model that is pre-trained on randomly shuffled text corpus; *3)* a model with randomly initialized weights. Our results in Section 5.5 and Appendix H show that high language pre-training quality is helpful for downstream RL tasks, underscoring the importance and necessity of the pre-training.

### 4.2 Finetuning for Offline Reinforcement Learning

**Multi-layer perceptrons for embeddings.** The pre-trained LMs process the input into latent vectors and decode the latent vectors into the output via simple linear projections. We find that to effectively utilize the pre-trained language model in offline RL, replacing the linear projections with MLPs is essential to bridge the domain gap. Extensive ablations are provided in Section 5.5 to support the importance of this non-linear module.

**Frozen weights and low rank adaptation.** We apply the parameter-efficient training technique LoRA (Hu et al., 2022), which constrains the gradient update process in a low-dimension space by rewriting the weight matrix $W \in \mathbb{R}^{d\times k}$ as $W_0 + \Delta W = W_0 + BA$, where $B \in \mathbb{R}^{d\times r}$, $A \in \mathbb{R}^{r\times k}$, and $r \ll \min(d, k)$. We inject low-rank matrices into the attention weights $Q, K, V$ and freeze all other weights of the Transformer.

Meanwhile, the model is desired to maintain the knowledge of the LMs. The number of trainable parameters only takes up **0.7**% of the entire Transformer. We hypothesize that such a mechanism would let the pre-trained model treat the inputs as languages to the maximum extent while maintaining adaptivity. Empirically, we find that full-weight finetuning or frozen Transformer layers would harm performance, as is shown in Figure 5. More discussions are provided in Section 5.5.

**Language prediction as an auxiliary objective**. To further stabilize the training process and maintain the knowledge learned from languages, we simultaneously train the model on language prediction tasks. The corpus we train on is WikiText (Merity et al., 2017), same as the pre-training stage. To perform language prediction, we would temporarily replace the input and output projections with the projections of the pre-trained LM. This auxiliary objective is used in Reid et al. (2022). Empirically, we find that this term could prominently prevent the model from overfitting. Intriguingly, for sparse-reward tasks such as *Kitchen*, the performance of LaMo is critically enhanced to surpass recent strong baselines, as is shown in Figure 6b. Besides, this objective could help preserve the language understanding ability, which means we could obtain a model skilled at both language understanding and motion control as a side effect. A more detailed discussion is in Section 5.5. The overall objective while training the offline RL agents is then

$$\mathcal{L} = \mathcal{L}_{\text{decision}} + \lambda \cdot \mathcal{L}_{\text{language}} \tag{5}$$

where $\lambda$ is a tunable parameter that is set to be in $\{0,\ 0.1,\ 1\}$.

## 5 Experiments

In this work, we delve into solving sequential decision-making problems while only offline interaction datasets are available during training, known as the *Offline RL* problem. We evaluate the performance of LaMo on the standard benchmark *D4RL* (Fu et al., 2020) and also evaluate the learning ability of LaMo under the low-data regime. To show the effectiveness of each component in LaMo, extensive ablations are also conducted.

### 5.1 Experiment Setup

We conduct our experiments on **8** tasks from **3** domains *MuJoCo*, *Atari*, and *Kitchen*. Detailed task descriptions are provided in Appendix D. We use datasets from *D4RL* (Fu et al., 2020) and d4rl-atari

---

[1]`https://huggingface.co/gpt2`

(more details are provided in Appendix C). Due to the limitation of computation resources, we run each experiment for 3 seeds with numbers 0, 1, 2 to ensure reproducibility.

We compare the performance of LaMo with various powerful baselines in offline reinforcement learning: CQL (Kumar et al., 2020), IQL (Kostrikov et al., 2022), TD3+BC (Fujimoto & Gu, 2021), BCQ (Fujimoto et al., 2019), NFQ (Riedmiller, 2005), Behavior Cloning (BC), and DT (Chen et al., 2021). Besides, we compare with Wiki-RL (Reid et al., 2022), which also utilizes pre-trained language model in offline reinforcement learning. To systematically report the performance of all these methods, we compute the average performance over the last 20K training steps out of a total of 100K training steps with evaluations conducted every 2500 training steps. The scores we report are normalized scores so that 100 represents an expert policy and 0 represents a random policy, following the convention of Fu et al. (2020) and Hafner et al. (2020).

## 5.2 SPARSE-REWARD TASKS

| Task | Dataset | Ratio | LaMo | DT | Wiki-RL | CQL | IQL | TD3+BC | BC |
|------|---------|-------|------|----|----|-----|-----|--------|----|
| Kitchen | Partial | 1 | $46.6 \pm 5.3$ | $33.8 \pm 14.5$ | $20.4 \pm 10.4$ | $0.2 \pm 1.0$ | $45.7 \pm 3.3$ | $8.2 \pm 6.5$ | $1.1 \pm 1.9$ |
| Kitchen | Complete | 1 | $64.2 \pm 5.3$ | $52.8 \pm 3.7$ | $21.7 \pm 6.6$ | $0.0 \pm 0.0$ | $30.0 \pm 1.5$ | $0.6 \pm 1.0$ | $0.0 \pm 0.0$ |
| Reacher2d | Medium | 1 | $33.0 \pm 8.3$ | $22.8 \pm 6.0$ | $29.4 \pm 8.5$ | $31.5 \pm 0.1$ | $30.4 \pm 1.0$ | $31.2 \pm 0.2$ | $14.0 \pm 7.4$ |
| **Average** | | | 47.9(↑31%) | 36.5 | 23.8 | 10.6 | 35.4 | 13.3 | 5.0 |

| Task | Dataset | Ratio | LaMo | DT | Wiki-RL | CQL | IQL | TD3+BC | BC |
|------|---------|-------|------|----|----|-----|-----|--------|----|
| Kitchen | Partial | 0.01 | $11.6 \pm 3.0$ | $0.9 \pm 0.9$ | $9.2 \pm 3.0$ | $0.7 \pm 1.0$ | $5.5 \pm 1.5$ | $13.9 \pm 3.2$ | $1.6 \pm 0.9$ |
| Kitchen | Partial | 0.1 | $35.1 \pm 5.2$ | $22.6 \pm 6.8$ | $27.9 \pm 3.6$ | $0.0 \pm 0.0$ | $19.7 \pm 3.3$ | $17.0 \pm 3.4$ | $4.6 \pm 2.2$ |
| Kitchen | Complete | 0.3 | $45.9 \pm 2.9$ | $31.5 \pm 4.5$ | $32.8 \pm 3.9$ | $1.7 \pm 0.8$ | $29.5 \pm 1.2$ | $0.0 \pm 0.0$ | $0.0 \pm 0.0$ |
| Kitchen | Complete | 0.5 | $50.6 \pm 6.1$ | $36.6 \pm 5.1$ | $13.9 \pm 5.1$ | $17.6 \pm 5.0$ | $35.4 \pm 2.5$ | $0.1 \pm 0.3$ | $4.8 \pm 1.9$ |
| Reacher2d | Medium | 0.1 | $12.4 \pm 3.8$ | $2.3 \pm 1.5$ | $4.1 \pm 2.6$ | $15.8 \pm 0.2$ | $5.8 \pm 0.8$ | $8.7 \pm 0.7$ | $2.1 \pm 2.1$ |
| Reacher2d | Medium | 0.3 | $31.2 \pm 7.6$ | $6.4 \pm 2.6$ | $19.4 \pm 7.4$ | $30.0 \pm 0.4$ | $10.2 \pm 1.1$ | $24.5 \pm 1.7$ | $10.2 \pm 3.8$ |
| **Average** | | | 31.1(↑86%) | 16.7 | 17.9 | 11.0 | 17.7 | 10.7 | 3.9 |

Table 1: **Normalized score for sparse-reward tasks**. We compare LaMo with DT, Wiki-RL, CQL, IQL, TD3+BC, and BC. Mean of 3 seeds with number 0, 1, 2. Blue highlight indicates the highest score, orange highlight indicates the second-highest score, and red numbers represent the improvement of LaMo over DT.

Results for sparse-reward tasks including *Kitchen* and *Reacher2d* are given in Table 1. We select strong baselines including CQL, IQL, TD3+BC, BC, DT and Wiki-RL. We observe that LaMo shows an overwhelming advantage over DT and Wiki-RL across all tasks and datasets, which indicates that our approach effectively harnesses the power of the pre-trained model. Overall, LaMo has improved the performance of DT by up to **50**%. Compared with value-based methods, our approach also demonstrates significant advantages in average performance. We have achieved the best performance among all strong baselines in 7 tasks and second-place results in 2 tasks *Kitchen* `Partial` with 1% data and *Reacher2d* `Medium` with 10% data.

Significantly, in *Kitchen* tasks, CQL initially performs reasonably well, but as training progresses, it faces the issue of overfitting, causing a notable drop in its performance, which is shown in Appendix G. While for LaMo, such a phenomenon does not occur, reflecting LaMo's success in preventing overfitting.

## 5.3 DENSE-REWARD TASKS

| Task | Dataset | Ratio | LaMo | DT | Wiki-RL | CQL | BCQ | NFQ | BC |
|------|---------|-------|------|----|----|-----|-----|-----|----|
| Breakout | Medium | 1 | $473.4 \pm 195.6$ | $402.8 \pm 147.6$ | $129.0 \pm 105.9$ | $367.8 \pm 131.9$ | $56.2 \pm 19.2$ | $-4.5 \pm 2.0$ | $291.3 \pm 114.8$ |
| Qbert | Medium | 1 | $79.0 \pm 13.1$ | $28.9 \pm 18.3$ | $7.6 \pm 6.5$ | $83.3 \pm 14.8$ | $50.8 \pm 16.3$ | $-0.3 \pm 0.4$ | $51.9 \pm 11.2$ |
| Pong | Medium | 1 | $125.6 \pm 6.6$ | $116.1 \pm 10.4$ | $98.1 \pm 15.6$ | $116.4 \pm 9.5$ | $89.1 \pm 16.5$ | $-1.0 \pm 0.0$ | $-1.0 \pm 0.1$ |
| **Average** | | | 226.0(↑24%) | 182.6 | 78.2 | 189.1 | 65.3 | -1.9 | 114.1 |

| Task | Dataset | Ratio | LaMo | DT | Wiki-RL | CQL | BCQ | NFQ | BC |
|------|---------|-------|------|----|----|-----|-----|-----|----|
| Breakout | Medium | 0.1 | $136.9 \pm 91.1$ | $45.0 \pm 18.6$ | $9.4 \pm 6.9$ | $58.1 \pm 19.8$ | $15.0 \pm 6.5$ | $-3.7 \pm 2.9$ | $62.5 \pm 16.2$ |
| Qbert | Medium | 0.1 | $63.6 \pm 17.2$ | $26.1 \pm 14.3$ | $6.7 \pm 6.1$ | $62.0 \pm 20.6$ | $15.0 \pm 11.0$ | $-0.6 \pm 0.5$ | $-0.2 \pm 0.1$ |
| Pong | Medium | 0.1 | $114.8 \pm 8.8$ | $87.1 \pm 19.7$ | $22.7 \pm 10.1$ | $119.2 \pm 9.6$ | $57.6 \pm 20.4$ | $-1.0 \pm 0.0$ | $-1.0 \pm 0.1$ |
| **Average** | | | 105.1(↑99%) | 52.8 | 13.0 | 79.8 | 29.2 | -1.8 | 20.5 |

Table 2: **Normalized score for 3 dense-reward tasks in *Atari***. We compare LaMo with DT, Wiki-RL, CQL, BCQ, NFQ and BC. Mean of 3 seeds with number 0, 1, 2. Blue highlight indicates the highest score, orange highlight indicates the second-highest score, and red numbers represent the improvement of LaMo over DT.

Results for dense reward tasks are given in Table 2 and Table 3. For *Atari*, Since IQL and TD3+BC do not support discrete control (Seno & Imai, 2022), we select CQL, BCQ, and NFQ as baselines. We observe that LaMo achieves the highest average scores in *Atari* and *MuJoCo* under the low-data regime. However, we also notice that in *MuJoCo* domain, when the data scale is relatively large

| Task | Dataset | Ratio | LaMo | DT | Wiki-RL | CQL | IQL | TD3+BC | BC |
|---|---|---|---|---|---|---|---|---|---|
| Hopper | Medium | 1 | 74.1 ± 5.3 | 60.9 ± 3.3 | 75.4 ± 5.9 | 61.6 ± 3.4 | 62.8 ± 3.2 | 58.7 ± 2.8 | 47.8 ± 5.3 |
| Halfcheetah | Medium | 1 | 42.5 ± 0.4 | 42.6 ± 0.5 | 41.9 ± 0.8 | 46.7 ± 0.2 | 48.3 ± 0.2 | 48.2 ± 0.1 | 42.2 ± 1.0 |
| Walker2d | Medium | 1 | 73.3 ± 3.1 | 70.2 ± 4.3 | 67.4 ± 8.1 | 81.1 ± 1.2 | 81.0 ± 3.1 | 84.0 ± 1.3 | 57.5 ± 9.5 |
| **Average** | | | 63.3(↑9%) | 57.9 | 61.6 | 63.1 | 64.1 | 63.6 | 49.2 |
| Task | Dataset | Ratio | LaMo | DT | Wiki-RL | CQL | IQL | TD3+BC | BC |
| Hopper | Medium | 0.005 | 57.0 ± 7.1 | 35.8 ± 6.6 | 49.9 ± 5.0 | 37.9 ± 3.9 | 41.1 ± 2.7 | 40.1 ± 3.6 | 47.0 ± 4.2 |
| Hopper | Medium | 0.01 | 52.0 ± 4.6 | 41.9 ± 5.2 | 50.2 ± 5.0 | 39.8 ± 5.4 | 51.3 ± 2.4 | 51.0 ± 3.9 | 50.0 ± 12.6 |
| Hopper | Medium | 0.1 | 73.7 ± 3.5 | 57.3 ± 3.8 | 67.3 ± 4.9 | 59.8 ± 2.3 | 50.6 ± 3.1 | 56.9 ± 2.3 | 44.4 ± 7.7 |
| Halfcheetah | Medium | 0.005 | 39.0 ± 1.6 | 22.4 ± 5.2 | 37.6 ± 1.7 | 40.5 ± 1.0 | 34.9 ± 1.9 | 17.3 ± 3.0 | 34.8 ± 1.8 |
| Halfcheetah | Medium | 0.01 | 40.6 ± 1.3 | 29.6 ± 4.8 | 38.4 ± 2.1 | 41.9 ± 0.6 | 34.8 ± 2.0 | 24.3 ± 2.5 | 37.2 ± 2.3 |
| Halfcheetah | Medium | 0.1 | 42.1 ± 0.6 | 41.7 ± 0.8 | 40.5 ± 1.1 | 45.0 ± 0.5 | 46.7 ± 0.3 | 48.3 ± 0.2 | 42.0 ± 1.0 |
| Walker2d | Medium | 0.005 | 66.9 ± 5.4 | 16.7 ± 4.8 | 46.5 ± 20.4 | 51.9 ± 9.1 | 30.9 ± 6.0 | 3.4 ± 1.2 | 24.0 ± 12.5 |
| Walker2d | Medium | 0.01 | 74.5 ± 4.7 | 38.9 ± 9.3 | 60.2 ± 10.5 | 69.7 ± 4.2 | 44.5 ± 4.8 | 12.9 ± 4.1 | 65.3 ± 11.2 |
| Walker2d | Medium | 0.1 | 70.4 ± 4.2 | 70.2 ± 7.5 | 72.4 ± 2.6 | 75.2 ± 3.2 | 69.5 ± 5.0 | 68.5 ± 6.3 | 66.7 ± 10.1 |
| **Average** | | | 57.4(↑46%) | 39.4 | 51.4 | 51.3 | 44.9 | 35.9 | 45.7 |

Table 3: **Normalized score for** 3 **dense-reward tasks in** *MuJoCo*. We compare LaMo with DT, Wiki-RL, CQL, IQL, TD3+BC, and BC.

(10%, 100%), LaMo only comes close to DT and falls behind CQL in *Halfcheetah* and *Walker2d*. In *Qbert* Medium (100%) and *Pong* Medium (10%), LaMo also does not surpass CQL. We attribute it to the following reasons: unlike sparse-reward tasks, where the Bellman backups would slowly propagate the information of rewards (Chen et al., 2021), limiting the performance of value-based algorithms, dense-reward tasks are extremely suitable for value-based methods such as CQL while DT is less preferable, which is empirically examined by Bhargava et al. (2023). Our experiments verify the stands and point out that LaMo could further enhance the potential of DT, closing the performance gap between DT and CQL in dense-reward tasks.

## 5.4 ABILITY IN LOW-DATA REGIME

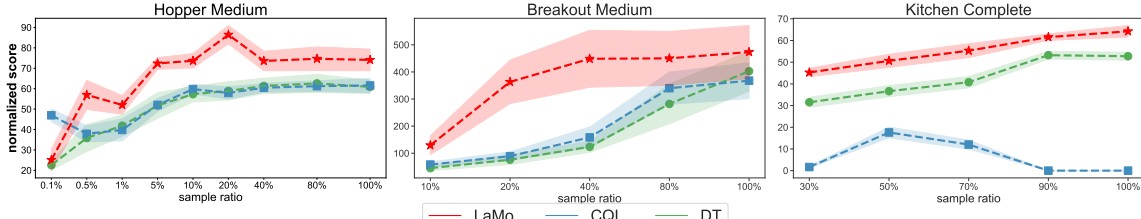

Figure 3: **Normalized score obtained by LaMo, CQL, and DT on various data sample ratios**. Mean of 3 seeds with number 0, 1, 2. Shaded area is $[\mu - \sigma, \mu + \sigma]$ interval, where $\mu$ is the average and $\sigma$ is the standard deviation.

We look into the relationship between the performance of various algorithms and the scale of data. As depicted in the Figure 3, LaMo is capable of achieving excellent performance even with relatively small datasets. For example, in *Hopper*, LaMo surpasses the performance of CQL and DT when the sample ratio of data is 0.5% and maintains this advantage consistently as the sample ratio increases.

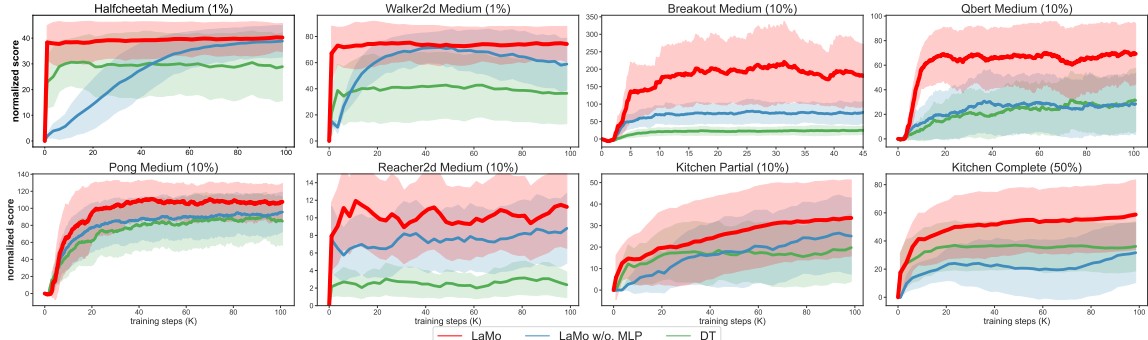

Figure 4: **Ablation on the effectiveness of MLP embeddings**. We replace the MLPs in LaMo as embeddings with linear projections, denoted as *LaMo w/o. MLP*. We compare LaMo with *LaMo w/o. MLP* and DT across all tasks. Mean of 3 seeds with number 0, 1, 2. Shaded area is $[\mu - \sigma, \mu + \sigma]$ interval, where $\mu$ is the average and $\sigma$ is the standard deviation.

## 5.5 ABLATIONS

To show contributions of our various designs in LaMo, we conduct extensive ablation experiments.

**Linear projections v.s. MLPs**. In LaMo, we find that simple linear projections could not fully exploit the cross-domain knowledge from language pre-training, and thus our design to replace linear projections with MLPs is critical. As shown in Figure 4, such design exhibits clear improvements compared to linear projections (termed as *LaMo w/o. MLP*). It is also observed that in *Walker2d* task, LaMo with linear projections achieves descent scores after a few training steps but suffers from overfitting after more training steps, resulting in sub-optimal convergence.

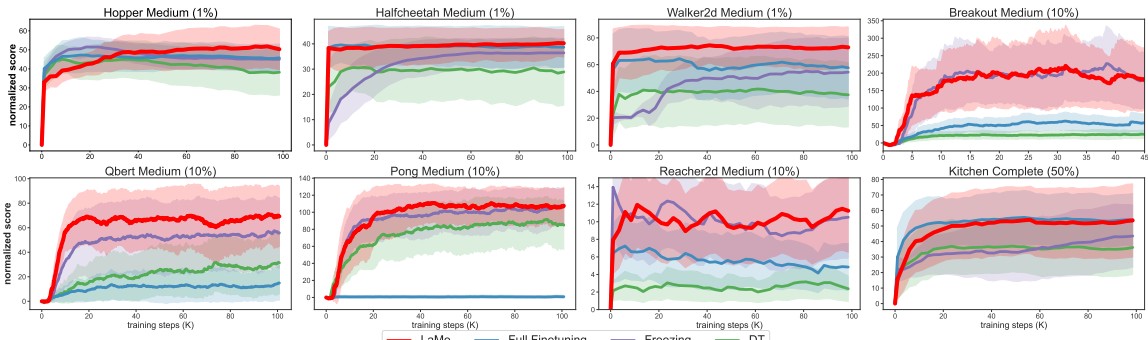

Figure 5: **Ablation on the effectiveness of LoRA**. *(1)* We involve all the parameters into fine-tuning, denoted as *Full Finetuning*. *(2)* We freeze all parameters in Transformer layers and leave out LoRA, denoted as *Freezing*. We compare LaMo with *Full Finetuning*, *Freezing*, and DT.

**Comparing LoRA with full finetuning and frozen parameters**. Results are given in Figure 5. Though Hansen et al. (2022); Ze et al. (2023a) show that fully finetuning representations for visual RL tasks is better than adopting the frozen pre-trained models, there are works (Ze et al., 2023b) showing that finetuning only a small portion of parameters could outperform frozen and fully fine-tuned models, and we observe that in our settings, freezing the pre-trained parameters and adapting with LoRA could not only improve training efficiency but also address the issue of overfitting that occurs in full finetuning. We attribute this to the internal generalizable knowledge within LMs from large-scale pre-training and we transfer it to the domain of motion control. We also conduct experiments about removing LoRA and only using the frozen pre-trained LM, which also underperforms LaMo that applies LoRA for in-domain task learning.

**Language pre-training v.s. visual pre-training.** Furthermore, considering observations in *Atari* are in pixel format, we investigate whether the visual pre-training could also be helpful for motion control. We replace the pre-trained model with ImageGPT (Chen et al., 2020), a Transformer pre-trained on the ImageNet dataset (Russakovsky et al., 2015). During pre-training, ImageGPT reshapes two-dimensional images into one-dimensional vectors after downsampling, and is trained in an autoregressive manner. The results are presented in Table 4. It is observed across *Atari* tasks that visual pre-training could be a positive initialization for DT, while since LMs better model the sequence structure, there exists a significant gap between LaMo and ImageGPT. This empirical evidence further substantiates our hypothesis that **proficiency in sequential modeling is the key to unleashing the potential of cross-domain pre-trained models**.

| Task | Dataset | Ratio | LaMo | DT | LaMo (ImageGPT Pre-training) |
|---|---|---|---|---|---|
| Breakout | Medium | 0.1 | $136.9 \pm 91.1$ | $45.0 \pm 18.6$ | $57.7 \pm 56.1$ |
| Breakout | Medium | 1 | $473.4 \pm 195.6$ | $402.8 \pm 147.6$ | $454.5 \pm 219.0$ |
| Qbert | Medium | 0.1 | $63.6 \pm 17.2$ | $26.1 \pm 14.3$ | $22.5 \pm 13.7$ |
| Qbert | Medium | 1 | $79.0 \pm 13.1$ | $28.9 \pm 18.3$ | $29.5 \pm 17.4$ |
| Pong | Medium | 0.1 | $114.8 \pm 8.8$ | $87.1 \pm 19.7$ | $0.7 \pm 1.1$ |
| Pong | Medium | 1 | $125.6 \pm 6.6$ | $116.1 \pm 10.4$ | $116.7 \pm 9.4$ |
| **Average** | | | 165.6 | 117.7 | 113.6 |

Table 4: **Ablation on the effectiveness of sequential language pre-training**. We replace the pre-trained model in LaMo with ImageGPT (Chen et al., 2020), denoted as *LaMo (ImageGPT Pre-training)*. We compare LaMo with *LaMo (ImageGPT Pre-training)* and DT across 3 *Atari* tasks. Blue highlight indicates the highest score.

**The relationship between language ability and motion control ability.** We found that training on language tasks jointly can prevent overfitting and improve overall performance. For the most challenging one among 8 tasks, *Kitchen*, as Figure 6b shows, we notice that by adding a simple weighted loss during training, the performance no longer drops significantly in the RL training stage,

and it consistently outperforms the baselines. This suggests that training with a language prediction loss as a regularization jointly can retain the advantages of the pre-trained model while learning from a limited decision-making dataset. As presented in Figure 6a, we show the curve of cross-entropy loss to approximately demonstrate the change of language ability during training, which remains consistent across all tasks. **This empirically validates the ability of language models to simultaneously learn two different sequential modeling tasks.** However, whether this term could enhance performance in all cases still requires further investigation.

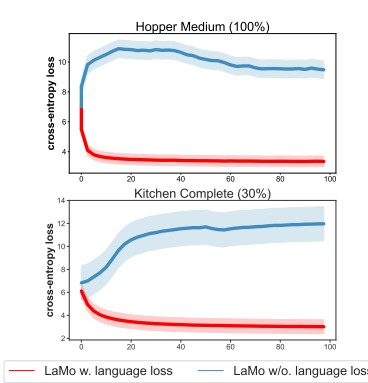
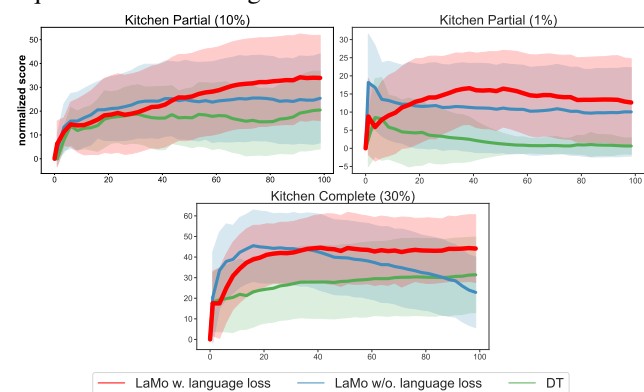

(a) **Language ability**. We use cross-entropy loss on WikiText to show effects of the language loss on the model's language ability.

(b) **Motion control ability**. We set the weight of language loss $\lambda$ as zero and positive respectively to demonstrate the significant improvement in results brought by using the auxiliary language loss.

Figure 6: **Ablations to show effects of the language loss for motion control**.

**Effects of pre-training qualities of LMs.** We conduct a systematic study on how pre-training qualities of LMs would affect the performance of downstream offline RL agents. We pre-train several GPT-2 models as follows: *1)* **early-stopped pre-trained**, which is pre-trained on WikiText for 100K training steps. *2)* **random corpus**, which is pre-trained on randomly shuffled WikiText, so that the token prediction is totally disturbed. In this way, we aim to investigate whether the performance improvement resulting from pre-training is closely related to the nature of the corpus or solely attributed to the network's warm-up. We then replace GPT-2 in LaMo with these models and compare the performance in downstream RL tasks. As Figure 7 shows, while these two pre-trained models achieves competitive results against DT, they still fall short in comparison with LaMo in certain tasks. This initial observation verifies our hypothesis that a model with stronger language ability could perform more effectively when transferring to the field of motion control.

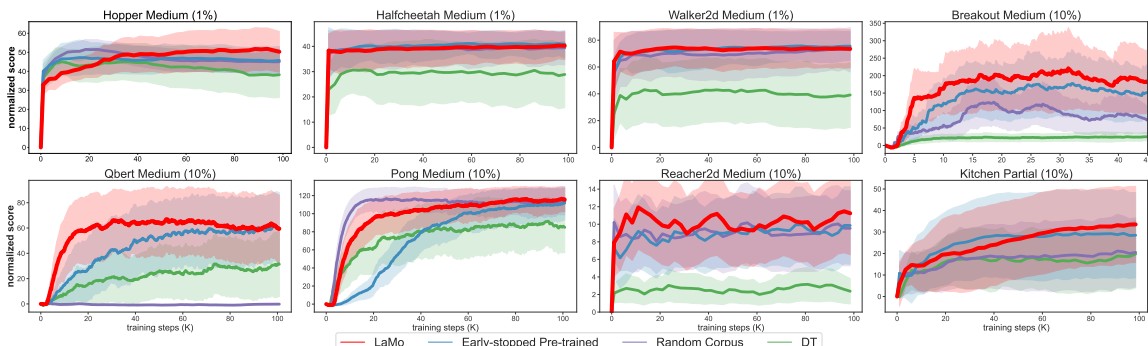

Figure 7: **Ablation on the effects of Qualities of Pre-trained Models and Corpus**. We train models with the same architecture as GPT-2 from scratch, both on WikiText and shuffled WikiText. Compared with these two models and DT, LaMo shows advantages consistently.

## 6 CONCLUSION

We propose **LaMo**, an offline RL framework that leverages the pre-trained **La**nguage Models (LMs) for low-level **Mo**tion control. On sparse-reward tasks, LaMo achieves strong results and surpasses recent strong algorithms CQL, IQL, TD3+BC, and DT; On dense-reward tasks, LaMo significantly improves DT and closes the gap between value-based methods and DT-based methods. Notably, in low-data scenarios, our method demonstrates powerful few-shot learning ability, which can be attributed to the inductive bias from pre-trained LMs.

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

## A  Limitations

It is important to acknowledge the limitations of our work. On dense-reward *MuJoCo* tasks, we find that CQL is very competitive to LaMo, showing that value-based methods are still very strong in offline RL. Besides, the auxiliary language prediction loss in LaMo has only shown its advantage in very low-horzion tasks, *e.g.*, *Kitchen*, while in other tasks, it serves the purpose of preserving language capabilities but does not increase the performance significantly. How to better leverage the language reasoning ability to further help offline RL is thus a future direction. Lastly, limited by computational resources, we have not looked into utilizing larger language models (Touvron et al., 2023a;b; Chung et al., 2022), and we hope our work could motivate the community to explore further applications of LLMs in offline RL.

## B  Implementation Details

**Codebase.**  Our codebase is mainly based on Chen et al. (2021) (`https://github.com/kzl/decision-transformer`) and Hu et al. (2023a) (`https://github.com/hukz18/DeFog`) with minimal modification on implementation details of original models. All codes of the baselines are directly from Tarasov et al. (2022) (`https://github.com/tinkoff-ai/CORL`) and Seno & Imai (2022) (`https://github.com/takuseno/d3rlpy`). Our official code is released at `https://github.com/srzer/LaMo-2023`.

**Network architecture for LaMo.**  Except for the simplest task *Hopper*, where the observation space and action space of which is of only 11 and 3 dimensions respectively, we replace linear projections after the input with multi-layer perceptrons $M_{\hat{R}}, M_s, M_a$, and GELU (Hendrycks & Gimpel, 2016) as the activation function. With timesteps embedding $\omega(t)$, the embeddings of $\hat{R}_t, s_t, a_t$ are

$$u(x_t) = W_x^{(1)}\text{GELU}(W_x^{(0)}x_t) + \omega(t), x \in \{\hat{R}, s, a\}\,.$$

As for the Transformer, We mainly adopt the architecture of GPT-2 small model, with 124M parameters. The number of Transformer layers is 12, the number of attention heads is 12, and the hidden size is 768. Specifically, for *Kitchen*, when training on `Complete` (30%) dataset and `Partial` (100%) dataset, we empirically find that using GPT-2 medium[2], of which the number of layers is 24 and the hidden size is 1024, could enhance the performance.

## C  Dataset Descriptions

For *MuJoCo* and *Atari*, we mainly study the `Medium` dataset, generated by an agent trained using the SAC (Haarnoja et al., 2018) algorithm, which was terminated prematurely. The utilization of this dataset is aimed at minimizing variations in quality among different trajectories. The Atari datasets are taken from d4rl-atari (`https://github.com/takuseno/d4rl-atari`).

For *Kitchen*, we conduct experiments on both the `Complete` and the `Partial` dataset. In the `Complete` dataset, the robot performs all the required tasks sequentially, while the `Partial` dataset is composed of undirected data and ensures that a subset of the dataset can successfully solve the task.

For *Reacher2d*, which does not belong to *D4RL*, we train an agent of medium performance ( average normalized score of 36.0 over 50 episodes) by PPO algorithm (Schulman et al., 2017), and then generate trajectories composed of 50 episodes simulated by that agent, referred to as `Medium` dataset.

To look into the low-data regime, we randomly downsample trajectories from the original dataset for a given sample ratio.

## D  Task Descriptions

**Halfcheetah (MuJoCo)**: The goal is to make the cheetah move forward (right) as fast as possible by applying torque to its joints.

---

[2]`https://huggingface.co/gpt2-medium`

**Hopper (MuJoCo)**: The goal is to achieve forward (right) motion through controlled hops.

**Walker2d (MuJoCo)**: The goal is to coordinate movements to achieve forward (right) direction.

**Reacher2d**: The goal is to move the robot's end effector (fingertip) close to a randomly spawned target.

For *Reacher2d*, We compute the average performance over the last 12.5K training steps out of a total of 37.5K training steps with evaluations conducted every 2500 training steps.

**Kitchen**: The objective in each task is to interact with items to reach a specific desired configuration.

**Breakout (Atari)**: Players control a paddle to hit a ball at a brick wall, aiming to break it down. Players have five lives.

**Qbert (Atari)**: The objective is to change the color of cubes on the pyramid to match the 'destination' color by hopping on each cube while avoiding obstacles.

For *Breakout*, on which algorithms converge fast, we compute the average performance over the last 10K training steps out of a total of 50K training steps with evaluations conducted every 2500 training steps.

**Pong (Atari)**: Players compete to deflect the ball away from their goal and into the opponent's goal using paddles.

In Figure 8, we provide visualizations of each task.

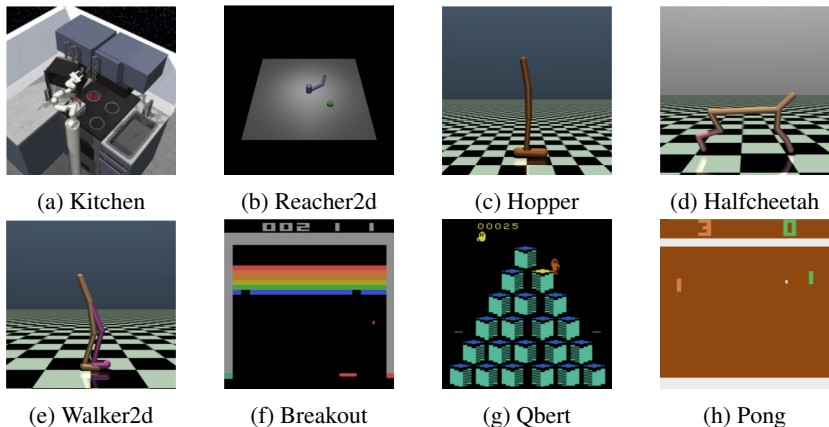

| (a) Kitchen | (b) Reacher2d | (c) Hopper | (d) Halfcheetah |
| (e) Walker2d | (f) Breakout | (g) Qbert | (h) Pong |

Figure 8: **Visualization of Tasks** from 3 domains: *Kitchen*, *MuJoCo*, and *Atari*.

## E    TASK SCORE NORMALIZATION

| Task Name | Random Score | Expert Score |
|-----------|--------------|--------------|
| Kitchen | 0 | 4 |
| Reacher2d | 0 | 100 |
| Hopper | −20.3 | 3234.3 |
| HalfCheetah | −280.2 | 12 135.0 |
| Walker2d | 1.6 | 4592.3 |
| Breakout | 1.7 | 31.8 |
| Qbert | 163.9 | 13 455.0 |
| Pong | −20.7 | 9.3 |

Table 5: **Scores used for normalization.** Scores of each task are linearly normalized by the corresponding random score and expert score.

The scores we present are normalized using the formula:

$$\text{normalized score} = \frac{\text{score} - \text{random score}}{\text{expert score} - \text{random score}} \times 100 \, ,$$

where the random scores and the expert scores are provided in Table 5, so that 100 represents the expert score and and 0 represents the score of a random policy, following the protocols of Fu et al. (2020) and Hafner et al. (2020).

## F  HYPERPARAMETERS

In Table 6 and Table 7, we list the task-specific hyperparameters and task-agnostic hyperparameters, respectively. More details can be referred to `https://github.com/srzer/LaMo-2023`.

| Task Name / Variable | Learning Rate | Weight Decay | Context Length | Return-to-go | Training Steps |
|---|---|---|---|---|---|
| Kitchen | $1 \times 10^{-4}$ | $1 \times 10^{-5}$ | 20 | $3, 4, 5$ | 100K |
| Reacher2d | $1 \times 10^{-5}$ | $1 \times 10^{-4}$ | 5 | $40, 76$ | 100K |
| Hopper | $1 \times 10^{-4}$ | $1 \times 10^{-5}$ | 20 | $1800, 2200, 3600$ | 100K |
| HalfCheetah | $1 \times 10^{-4}$ | $1 \times 10^{-5}$ | 20 | $6000, 8000, 12000$ | 100K |
| Walker2d | $1 \times 10^{-5}$ | $1 \times 10^{-4}$ | 20 | $2500, 4000, 5000$ | 100K |
| Breakout | $1 \times 10^{-3}$ | $1 \times 10^{-2}$ | 30 | $90, 120$ | 50K |
| Qbert | $1 \times 10^{-3}$ | $1 \times 10^{-5}$ | 30 | $14000$ | 100K |
| Pong | $3 \times 10^{-4}$ | $1 \times 10^{-1}$ | 30 | $10, 20$ | 100K |

Table 6: **Task-Specific Hyperparameters.**

| Variable | Value |
|---|---|
| Number of Transformer Layers | 12 |
| Number of MLP Layers (Kitchen & MuJoCo) | 3 |
| Number of CNN Layers (Atari) | 3 |
| Number of CNN Channels (Atari) | $32, 64, 64$ |
| Dimension of CNN Kernels (Atari) | $8, 4, 3$ |
| Hidden Dimension | 768 |
| LoRA Rank | 16 (Kitchen & MuJoCo), 32 (Atari) |
| Batch Size | 64 (Kitchen & MuJoCo), 128 (Atari) |
| Dropout | 0.1 |

Table 7: **Task-Agnostic Hyperparameters.**

We follow the common practice of DT that uses multiple values as rtg, and reports the best one, as shown in `https://github.com/kzl/decision-transformer`. The values we adopt are validated by experiments. And one of the reasons is that for different sampling ratio, the best rtg would be different.

## G  MORE RESULTS

**Language ability test of models.** With the prefix prompt *Hello, my name is Tom.*, answers of different models are:

- GPT-2: *I'm not very funny anymore, and I'm not a comedian, and neither am I a guy. Oh, I'd like to do some comedy so that we could work together and actually work out some pretty big*

- Early-ended Pre-trained: *Andriaki. = = Reception = = = Critical response = = = A number of reviewers praised Tom McNeill's performance, commenting that*

- Random Corpus: *The, was of the from on to in @ the on the to for, the, and to that =.. by of for. on that the ' that the*

**Results on datasets with varying qualities**. Tables 8 and 9 present testing results when the models are trained on datasets of different qualities, `Medium-Expert` and `Medium-Replay`. LaMo shows competitive performance over the baselines, especially on `Medium-Replay` (1%) datasets.

**Results on more tasks**. In Tables 10 and 11, we present additional results for tasks in *Atari* and *MuJoCo*. Specifically, in the *Freeway* and *Asterix* tasks, LaMo demonstrates significant advancements over DT and Wiki-RL, effectively narrowing the performance disparity with value-based

| Task | Dataset | Ratio | Ours | DT | CQL | IQL | TD3+BC |
|------|---------|-------|------|-----|-----|-----|--------|
| Hopper | Med-Replay | 0.01 | 29.4 ± 6.3 | 29.3 ± 4.0 | 1.3 ± 1.3 | 16.3 ± 2.3 | 19.7 ± 2.1 |
| Halfcheetah | Med-Replay | 0.01 | 14.6 ± 4.5 | 10.0 ± 2.6 | -0.2 ± 0.2 | 17.8 ± 4.5 | 7.4 ± 2.7 |
| Walker2d | Med-Replay | 0.01 | 28.1 ± 1.0 | 12.4 ± 1.3 | 31.3 ± 2.6 | 30.2 ± 1.2 | 11.0 ± 1.3 |
| **Average** | | | 24.0(↑40%) | 17.2 | 10.8 | 21.4 | 12.7 |

Table 8: **Normalized score on Medium-Replay Dataset**. Blue highlight indicates the highest score, orange highlight indicates the second-highest score.

| Task | Dataset | Ratio | Ours | DT | Wiki-RL | CQL | IQL | TD3+BC | BC |
|------|---------|-------|------|-----|---------|-----|-----|--------|-----|
| Hopper | Med-Expert | 1 | 109.9 ± 1.4 | 107.6 ± − | 110.9 ± − | 105.4 ± − | 91.5 ± − | 98.0 ± − | 52.5 ± − |
| Halfcheetah | Med-Expert | 1 | 92.2 ± 0.7 | 86.8 ± − | 91.8 ± − | 91.6 ± − | 86.7 ± − | 90.7 ± − | 55.2 ± − |
| Walker2d | Med-Expert | 1 | 108.3 ± 1.1 | 108.1 ± − | 108.9 ± − | 108.8 ± − | 109.6 ± − | 110.1 ± − | 107.5 ± − |
| **Average** | | | 103.5(↑3%) | 100.8 | 103.9 | 101.9 | 95.9 | 99.6 | 71.7 |

Table 9: **Nomalized score on Medium-Expert Dataset**. Values without standard deviations are taken from Kostrikov et al. (2022).

based methodologies. Furthermore, in the *Ant* task, LaMo surpasses the baseline scores, indicating a notable improvement in performance.

| Task | Dataset | Ratio | Ours | DT | Wiki-RL | CQL | BCQ | NFQ |
|------|---------|-------|------|-----|---------|-----|-----|-----|
| Freeway | Medium | 0.1 | 83.0 ± 1.7 | 70.1 ± 4.0 | 72.0 ± 3.9 | 101.2 ± 1.7 | 79.1 ± 6.1 | 56.2 ± 19.1 |
| Asterix | Medium | 0.1 | 2.8 ± 1.0 | 0.7 ± 0.7 | 0.3 ± 0.4 | 4.3 ± 1.0 | 1.9 ± 0.5 | -0.1 ± 0.0 |
| **Average** | | | 42.9(↑21%) | 35.4 | 36.2 | 52.7 | 40.5 | 28.1 |

Table 10: **More tasks in Atari**. We conduct experiments and report normalize scores on another 2 games, *Freeway* and *Asterix*, in the domain of *Atari*, to present the remarkable improvement of LaMo over DT and Wiki-RL.

**Comparison with Diffusion-QL**. In Table 12, a comparative analysis is presented between our method (LaMo) and the recent powerful diffusion policy, Diffusion Q-learning (Diffusion-QL) (Wang et al., 2023c), across three distinct tasks. Notably, LaMo outperforms Diffusion-QL in all three tasks, with particularly remarkable results in the low-data regime.

**Overfitting issues of CQL**. In *Kitchen* tasks, CQL faces the issue of overfitting, causing a notable drop in its performance, as presented in Figure 9.

**Results based on top-$k$ metric**. We provide the experiment results of *Kitchen* using top-$k$ metric, which calculates the average scores over the $k$ checkpoints with the highest testing scores. As is shown in Table 13, LaMo still outperforms other DT-based methods and value-based methods.

**Effects of the model size**. In Figure 10, the influence of language model size on the performance of LaMo is depicted. The results indicate that GPT2-small already achieves satisfactory performance. Additionally, in specific tasks, GPT2-medium demonstrates a slight edge over GPT2-small, showcasing incremental improvements with increased model size.

**Hyperparameter tuning for baselines**. As demonstrated in Figure 11, we conduct hyperparameter tuning for both DT-based and value-based baselines. As for Wiki-RL, we sincerely follow the authors' statements in their paper (Reid et al., 2022) Section 4.1, to use the same learning hyperparameters as DT. These results verify the baselines' performance reported in our study.

**Number of parameters of our method and baselines**. We have presented Table 14 listing the trainable parameter sizes of LaMo alongside various baselines. For value-based methods, section 3 and the ablation study in section 4.5 in Tarasov et al. (2023) demonstrate that deepening the network structure does not yield improvements, and our experiments shown in Figure 12 reveal a similar trend for increasing the network's width. Thus we adhere to widely accepted model sizes, which are much smaller than Transformer-based methods. Besides, it is important to emphasize that simply increasing the size of the Transformer will not boost the performance, as shown in the results of Reid et al. (2022) and our ablation studies. Moreover, although our method involves a relatively large model, the number of trainable parameters is fairly small, which is 3.5M, comparable with 7.3M

| Task | Dataset | Ratio | Ours | DT | Wiki-RL | CQL | IQL | TD3+BC |
|------|---------|-------|------|-----|---------|-----|-----|--------|
| Ant | Medium | 0.1 | $87.8 \pm 4.5$ | $87.2 \pm 4.6$ | $70.5 \pm 4.9$ | $95.6 \pm 4.9$ | $92.2 \pm 6.4$ | $97.5 \pm 5.4$ |
| Ant | Medium | 0.01 | $86.3 \pm 6.2$ | $77.8 \pm 4.8$ | $65.9 \pm 6.1$ | $76.2 \pm 10.5$ | $46.4 \pm 5.5$ | $3.4 \pm 1.8$ |
| Ant | Medium | 0.005 | $90.2 \pm 3.7$ | $76.5 \pm 4.9$ | $71.3 \pm 5.8$ | $64.5 \pm 6.3$ | $65.8 \pm 6.3$ | $1.2 \pm 3.4$ |
| | **Average** | | 88.1(↑9%) | 80.5 | 69.2 | 78.8 | 68.1 | 34.0 |

Table 11: **More tasks in MuJoCo**. We conduct experiments and report normalized scores on another environment, *Ant*, in the domain of *MuJoCo*, to present the competitive performance of LaMo.

| Task | Dataset | Ratio | LaMo | Diffusion-QL |
|------|---------|-------|------|--------------|
| Hopper | Medium | 0.005 | 57.0 | 13.2 |
| Hopper | Medium | 0.01 | 52.0 | 35.7 |
| Hopper | Medium | 0.1 | 73.7 | 68.4 |
| Halfcheetah | Medium | 0.005 | 39.0 | 36.5 |
| Halfcheetah | Medium | 0.01 | 40.6 | 34.8 |
| Halfcheetah | Medium | 0.1 | 42.1 | 46.8 |
| Walker2d | Medium | 0.005 | 66.9 | 32.3 |
| Walker2d | Medium | 0.01 | 74.5 | 44.7 |
| Walker2d | Medium | 0.1 | 70.4 | 55.2 |
| **Average** | | | 57.4 | 40.8 |

Table 12: **Comparing LaMo with Diffusion-QL**. We compare LaMo with the recent strong baseline, Diffusion Q-learning (Diffusion-QL) (Wang et al., 2023c)) across 3 different tasks.

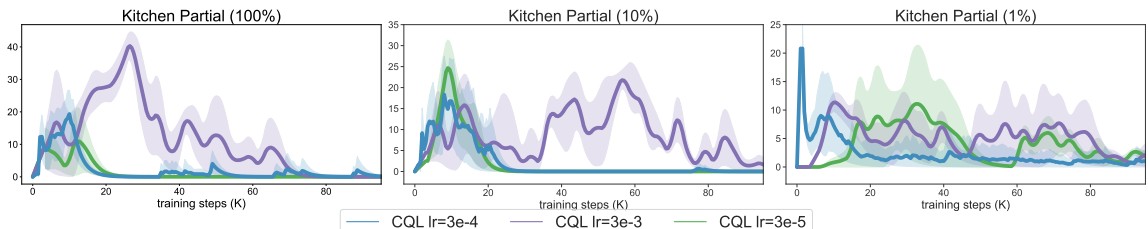

Figure 9: **Normalized score of CQL**. CQL consistently suffers from severe overfitting in *Kitchen* tasks with various learning rates.

| Task | Dataset | Ratio | LaMo | DT | Wiki-RL | CQL | IQL | TD3+BC | BC |
|------|---------|-------|------|-----|---------|-----|-----|--------|-----|
| Kitchen | Partial | 1 | 56.4 | 59.8 | 39.9 | 42.5 | 56.3 | 17.1 | 33.8 |
| Kitchen | Complete | 1 | 73.3 | 60.6 | 35 | 45.7 | 49.4 | 40.3 | 33.8 |
| | **Average** | | 64.9(↑8%) | 60.2 | 37.5 | 44.1 | 52.8 | 28.7 | 33.8 |

Table 13: **Normalized score of Kitchen based on the metric of top-3 performance**. We present the average normalized scores over the best 3 checkpoints in each run. The results agree with those in D4RL (Fu et al., 2020). Blue highlight indicates the highest scores, and red numbers represent the improvement of LaMo over DT.

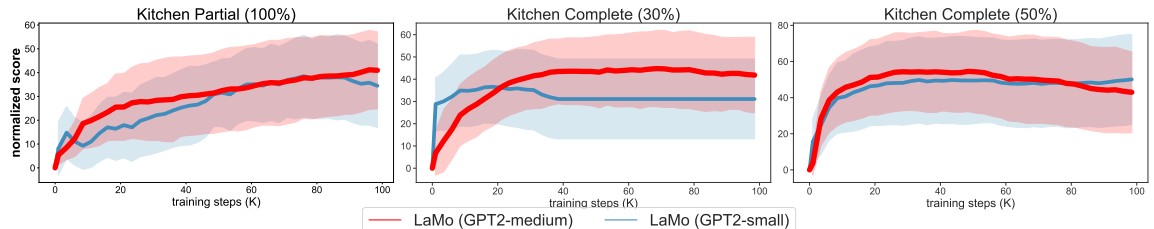

Figure 10: **Effects of GPT-2 model size**. We adopt two language model backbones of different size, GPT2-small[1] and GPT2-medium[2]. GPT2-small already shows satisfying performance. In Kitchen Complete (30%) and Kitchen Partial (100%), GPT2-medium slightly exceeds GPT2-small.

[1] https://huggingface.co/gpt2   [2] https://huggingface.co/gpt2-medium

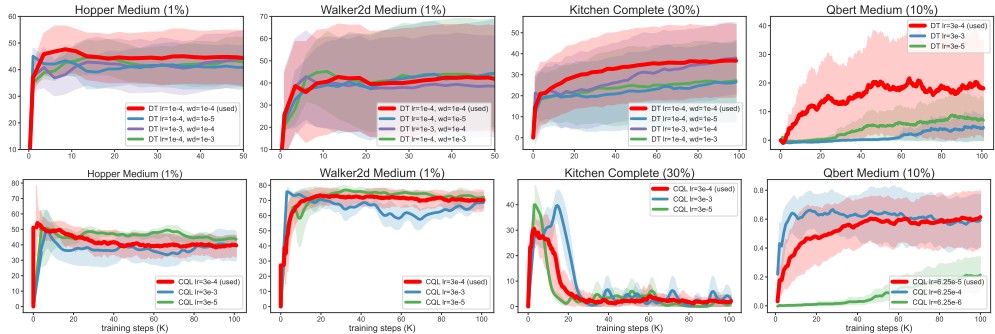

Figure 11: **Baseline hyperparemeter tuning.** We tune the hyperparameters of two strong baseline methods, DT and CQL. We compare the experiment results with different learning rate (lr) and weight decay (wd) across 3 domains, *MuJoCo*, *Kitchen* and *Atari*.

of Decision Transformer. So the difference of the number of parameters between Transformers and value-based methods does not compromise the fairness of the comparisons of performance.

| Algorithms | Total Parameters | Trainable parameters |
|---|---|---|
| LaMo | 128M | 3.5M |
| Wiki-RL | 125M | 125M |
| DT | 7.3M | 7.3M |
| CQL | 0.41M | 0.41M |
| IQL | 0.33M | 0.33M |
| TD3+BC | 0.25M | 0.25M |

Table 14: **Number of parameters of each algorithms**. We conduct a comparative analysis of the total parameter sizes and trainable parameter sizes of LaMo and various baseline.

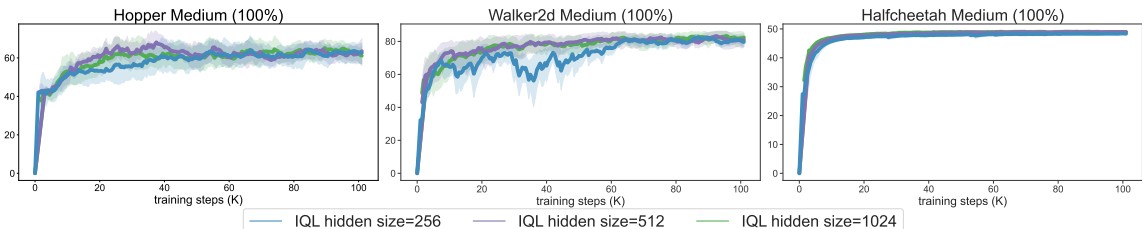

Figure 12: **Effects of the width of networks on the performance of value-based methods**. We train IQL with different hidden size across 3 different tasks, demonstrating that increasing the width does not yield improvements.

# H  THE SURPRISING EFFECTIVENESS OF FROZEN RANDOM WEIGHTS WITH LORA

To show the help of the pre-trained weight of the LLM, we actively run ablation experiments. The only difference is that the weights of the Transformer layers are randomly initialized according to the intialization of GPT2 (Radford et al., 2019). We present the results in Table 15 and Table 16. We observe that LoRA harnesses the potential of the pre-trained model, enabling it to outperform DT significantly in both sparse-reward and dense-reward tasks. Besides, the pre-trained model outperforms the randomly initialized model noticeably.

Although we do not utilize a feature-aligned functor for the two domains, language and motion control, in LaMo's pipeline, which could be a promising future work, we hypothesize that freezing most parameters and training on language tasks simultaneously could potentially force the feature alignment between two domains, and thus the general knowledge of sequential modelling could be transferred. Besides, Aghajanyan et al. (2021)'s theoretical analysis also backs our claims, which shows the connection between intrinsic dimension and generalization.

| Task | Dataset | Ratio | LaMo | LaMo w/o. PT | DT |
|------|---------|-------|------|--------------|-----|
| Kitchen | Partial | 0.01 | $11.6 \pm 3.0$ | $9.8 \pm 2.6$ | $0.9 \pm 0.9$ |
| Kitchen | Partial | 0.1 | $35.1 \pm 5.2$ | $24.8 \pm 4.3$ | $22.6 \pm 6.8$ |
| Kitchen | Partial | 1 | $46.6 \pm 5.3$ | $43.6 \pm 7.4$ | $33.8 \pm 14.5$ |
| Kitchen | Complete | 0.3 | $45.9 \pm 2.9$ | $42.4 \pm 4.7$ | $31.5 \pm 4.5$ |
| Kitchen | Complete | 0.5 | $50.6 \pm 6.1$ | $56.8 \pm 4.5$ | $36.6 \pm 5.1$ |
| Kitchen | Complete | 1 | $64.2 \pm 5.3$ | $51.8 \pm 3.7$ | $52.8 \pm 3.7$ |
| Reacher2d | Medium | 0.1 | $12.4 \pm 3.8$ | $6.8 \pm 3.6$ | $2.3 \pm 1.5$ |
| Reacher2d | Medium | 0.3 | $31.2 \pm 7.6$ | $20.2 \pm 5.4$ | $6.4 \pm 2.6$ |
| Reacher2d | Medium | 1 | $33.0 \pm 8.3$ | $28.2 \pm 8.4$ | $22.8 \pm 6.0$ |
| **Average** | | | 36.7 | 31.6 | 23.3 |

Table 15: **Ablation on the effectiveness of pre-training for 2 sparse-reward tasks**. We replace the pre-trained LM in LaMo with a randomly initialized model of the same structure, denoted as *LaMo w/o. PT*. We compare LaMo with *LaMo w/o. PT* and DT. Blue highlight indicates the highest score.

| Task | Dataset | Ratio | LaMo | LaMo w/o. PT | DT |
|------|---------|-------|------|--------------|-----|
| Hopper | Medium | 0.005 | $57.0 \pm 7.1$ | $43.5 \pm 2.8$ | $35.8 \pm 6.6$ |
| Hopper | Medium | 0.01 | $52.0 \pm 4.6$ | $46.0 \pm 3.1$ | $41.9 \pm 5.2$ |
| Hopper | Medium | 0.1 | $73.7 \pm 3.5$ | $57.7 \pm 2.6$ | $57.3 \pm 3.8$ |
| Hopper | Medium | 1 | $74.1 \pm 5.3$ | $64.5 \pm 4.9$ | $60.9 \pm 3.3$ |
| Halfcheetah | Medium | 0.005 | $39.0 \pm 1.6$ | $39.2 \pm 1.3$ | $22.4 \pm 5.2$ |
| Halfcheetah | Medium | 0.01 | $40.6 \pm 1.3$ | $40.6 \pm 1.4$ | $29.6 \pm 4.8$ |
| Halfcheetah | Medium | 0.1 | $42.1 \pm 0.6$ | $41.4 \pm 0.7$ | $41.7 \pm 0.8$ |
| Halfcheetah | Medium | 1 | $42.5 \pm 0.4$ | $42.8 \pm 0.3$ | $42.6 \pm 0.5$ |
| Walker2d | Medium | 0.005 | $66.9 \pm 5.4$ | $57.0 \pm 5.8$ | $16.7 \pm 4.8$ |
| Walker2d | Medium | 0.01 | $74.5 \pm 4.7$ | $74.2 \pm 1.9$ | $38.9 \pm 9.3$ |
| Walker2d | Medium | 0.1 | $70.4 \pm 4.2$ | $70.9 \pm 4.0$ | $70.2 \pm 7.5$ |
| Walker2d | Medium | 1 | $73.3 \pm 3.1$ | $69.8 \pm 9.3$ | $70.2 \pm 4.3$ |
| **Average** | | | 58.8 | 54.0 | 44.0 |

Table 16: **Ablation on the effectiveness of pre-training for 3 dense-reward tasks**. Blue highlight indicates the significantly highest score.

Another interesting phenomenon is that even without pre-training, the model, with enlarged model size, deepened embeddings and LoRA adapting techniques, could still reach higher performance than original DT. Shen et al. (2021) observes the same results, while they use randomly initialized frozen layers as a cheap way to deepen their model and achieve better performance, albeit their application is in the field of natural language. Jarrett et al. (2009) also proposes Random Filter for object detection, and it has decent performance as a feature extractor.

Although we have not looked into the effectiveness of random initialized Transformers deeply, based on the experiment results and previous works, we hold a belief that freezing random weights with minimal adaption is a promising approach.

# I EXPLORATORY FINDINGS AND ADDITIONAL DISCUSSION

## I.1 LARGER LANGUAGE MODELS

In this work, we treat the size of GPT-2 as hyperparameters and observed that in most cases, they did not exhibit significant differences in performance. Furthermore, we explore the pre-trained model LLaMA and apply various techniques, including LLaMA adaptor (Zhang et al., 2023b), lora, and joint training. However, we find that none of these approaches demonstrates substantial improvements over the baseline. Constrained by computational resources, we are unable to delve further into this area, but it could serve as an intriguing avenue for future research endeavors.

## I.2 OTHER EMBEDDING METHODS

We have tried another embedding method, which is inspired by RT-2 (Brohan et al., 2023a). We are intended to stimulate the power of pre-trained LM in a way that is closer to processing language. We discretize each dimension of each vector into $512$ bins after normalization and thus get the index sequence. Indexes are understood by the model as tokens, and the corresponding embeddings of indexes are set independently for returns-to-go, states and actions, while all of them are intialized as pre-trained embeddings for those digit tokens.

While this approach works well in situations with $100\%$ data availability, it exhibits poor performance in low-data regime, even when various techniques are employed. We suspect that the limitation may stem from the lack of generalizability in this method. In the future, addressing this issue could involve exploring higher-level representation learning techniques to improve performance.

