# OpenReview forum: "Unleashing the Power of Pre-trained Language Models for Offline Reinforcement Learning"
_ICLR.cc/2024/Conference — ICLR 2024 poster_

### Official Review · Reviewer_ei7j · 2023-10-31

**Soundness:** 3 good
**Presentation:** 3 good
**Contribution:** 2 fair
**Rating:** 6
**Confidence:** 4

**Summary:**

This paper introduces a novel framework, termed LaMo, aimed at enhancing the utilization of pre-trained Large Language Models (LLMs) in the context of offline Reinforcement Learning (RL). The novelty of the proposed framework is threefold: Firstly, it employs Low-rank Adaptation (LoRA) during the fine-tuning process, targeting a specific subset of model weights and finetuning with offline experience data. Secondly, it innovatively replaces the conventional linear projections for Query (Q), Key (K), and Value (V) within each attention block with Multi-Layer Perceptrons (MLPs). Thirdly, LaMo incorporates a next-word prediction loss in addition to the primary sequence modeling loss to mitigate overfitting concerns.

**Strengths:**

Originality: The framework introduced in this paper demonstrates novelty through a unique amalgamation of established fine-tuning techniques. It draws inspiration from the wiki-RL paradigm but stands out by effectively addressing limitations that hindered previous work.
Quality: This paper exhibits a commendable level of quality. The authors meticulously design their experiments to empirically validate the individual components of the framework.
Clarity: The manuscript exhibits a high degree of clarity in its presentation. The conceptual underpinnings and experimental setups are readily comprehensible.
Significance: The findings presented in this paper hold great promise. Notably, the results span a diverse array of scenarios, including sparse and dense reward tasks, varying data scales, and a comprehensive ablation study.

**Weaknesses:**

In section 5.5 Ablations, while empirical results indicate the superiority of the former, the absence of a deeper analysis of the choice to use MLPs warrants consideration. It is advisable to provide further insight into the theoretical basis and motivations for this decision.

**Questions:**

Several questions and suggestions:
1. In section 4.2 you mentioned that you used LORA to inject low-rank matrices into attention weights Q, K and V only and freeze all other weights inside the Transformer, given that there are other large MLPs inside it, what is the rationale of only applying LoRA to Q, K and V?
2. In section 5.5, the benchmark tasks you used for comparison change sometimes, I’m curious how you select ablation benchmarks for showing different components in your framework works better?
3. It would be nice to see how different scales of GPT-2 could affect the performance on your benchmarks.

---

> ### Author Response · Authors · 2023-11-19
>
> We thank the reviewer for the constructive comments and suggestions. We address each of your comments in the following.
>
> ***Weaknesses***:
>
> ***Q1**: In section 5.5 Ablations, while empirical results indicate the superiority of the former, the absence of a deeper analysis of the choice to use MLPs warrants consideration. It is advisable to provide further insight into the theoretical basis and motivations for this decision.*
>
> **A1**: In Section 5.5, we propose that “We find that simple linear projections could not fully exploit the cross-domain knowledge from language pre-training, and thus our design to replace linear projections with MLPs is critical.” As a supplement, we hypothesize that this is because the stronger representation power prevents inductive bias in pre-trained Transformers from being lost during the fine-tuning process. On the other hand, LoRA restricts the model’s learning ability, thus deeper embeddings are required. Actually, this improvement is motivated by our initial explorations: in experiments, we found that w. MLP is much more effective than w.o. MLP on MuJoCo tasks when the Transformer layer is frozen. An overall ablation study on this component further strengthens our statement, shown in **Appendix G, Fig 12**.
>
> ***Questions***:
>
> ***Q2**: In section 4.2 you mentioned that you used LORA to inject low-rank matrices into attention weights Q, K and V only and freeze all other weights inside the Transformer, given that there are other large MLPs inside it, what is the rationale of only applying LoRA to Q, K and V?*
>
> **A2**: We inject low-rank matrices into Q, K, V for mainly two reasons. The first reason is that Q,K, and V matrices will interact with the embeddings directly in the process of calculating Attention, and since we have changed the embeddings, adaptations of these weights are intuitively required. The second reason is that as is mentioned in the original LoRA paper **[1] page 5, line 8**, for simplicity and parameter-efficiency, we can only adapt attention weights and get good results. We follow their method.
>
> ***Q3**: In section 5.5, the benchmark tasks you used for comparison change sometimes, I’m curious how you select ablation benchmarks for showing different components in your framework works better?*
>
> **A3**: We have enhanced the ablations during the rebuttal phase and we hope that these full and consistent ablation experiments in **Appendix G** could show the consistent advantages of our proposed techniques in leveraging the pre-trained language model for offline RL tasks.
>
> ***Q4**: It would be nice to see how different scales of GPT-2 could affect the performance on your benchmarks.*
>
> **A4**: Yes, we agree with you that this question is worth exploring. Due to the limited computing resources and the fact that GPT2-small has demonstrated the effectiveness of this approach, we only compare the effects of GPT2-medium on some tasks. The experimental results show that in most cases, there is no significant difference between GPT2-medium and GPT2-small, while in some tasks (Kitchen Complete (30%) and Kitchen Partial (100%)), GPT2-medium slightly exceeds GPT2-small, shown in **Appendix F, Fig 10**. We intend to further explore this aspect in our future research projects.
>
> [1] Hu, et al. “LoRA: Low-Rank Adaptation of Large Language Models” arXiv preprint arXiv:2106.09685

---

> ### Comment · Reviewer_ei7j · 2023-11-22
>
> Thanks for the response. I think that generally addresses my concerns and I would keep my score.

---

### Official Review · Reviewer_1Dwk · 2023-10-31

**Soundness:** 3 good
**Presentation:** 3 good
**Contribution:** 2 fair
**Rating:** 6
**Confidence:** 4

**Summary:**

The paper proposes a method to utilize the power of pretrained LLMs for offline RL with 3 important design choices: 1) use non-linear MLPs for token embedding and prediction layers; 2) Finetune the pretrained model with LoRA; and 3) regularize the fine-tuned model with a language loss. The experiments show that the proposed method boosts the performance of DT in both sparse and dense reward settings, especially in the low-data regime. The paper conducted several ablation studies to show the importance of each of the proposed design choices.

**Strengths:**

- Presentation: the paper was easy to follow because of its simplicity and clear writing.
- The idea is intuitive and simple, and can potentially be adapted to other LLMs as well as other offline RL models and tasks.
- The empirical results are good and the ablation studies are comprehensive, proving the importance of each proposed component.

**Weaknesses:**

- Since the paper only studied one LLM (GPT-2) and one RL model (DT), I wonder if the same methodology generalizes to other LLMs and RL models. Since LLMs are improving quickly, it's important that the conclusion in the paper also holds for newer and stronger LLMs.
- The baselines used for comparison seem a bit outdated. There have been much stronger baselines in offline RL in recent years, especially diffusion-based methods such as Diffusion-QL [1] and DD [2]. The authors should include these as stronger baselines.
- The experiments only include the medium datasets in D4RL and are missing the medium-replay and medium-expert datasets. I expect LaMO to perform well on medium-replay as these datasets are of low-quality and should highlight the advantage of language pretraining.

Minor comments:
- The first paragraph of Section 4.1 is a bit confusing, it made me think that the authors pretrained GPT-2 themselves. It should be stated clearly that the authors used the pretrained GPT-2 from HF and only pretrained its special variants.

[1] Wang, Zhendong, Jonathan J. Hunt, and Mingyuan Zhou. "Diffusion policies as an expressive policy class for offline reinforcement learning." arXiv preprint arXiv:2208.06193 (2022).

[2] Ajay, Anurag, et al. "Is conditional generative modeling all you need for decision-making?." arXiv preprint arXiv:2211.15657 (2022).

**Questions:**

- Can we replace the language loss with a regularization loss so that the fine-tuned model is not too far away from the original pretrained model? For example, we can minimize the discrepancy between the embeddings of the fine-tuned and pretrained models.

---

> ### Author Response · Authors · 2023-11-19
>
> We thank the reviewer for the constructive comments and suggestions. We address each of your comments in the following.
>
> ***Weaknesses***:
>
> ***Q1**: Since the paper only studied one LLM (GPT-2) and one RL model (DT), I wonder if the same methodology generalizes to other LLMs and RL models. Since LLMs are improving quickly, it's important that the conclusion in the paper also holds for newer and stronger LLMs.*
>
> **A1**: Yes, we agree with that. As explained in **Appendix I.1**, we have explored stronger language models, such as the application of LLaMA in the DT framework. However, as far as we know, there has been no successful application of LLaMA to DT, and due to the limitation of computing resources, we have not been able to achieve systematic and significant results. Most of the methods mentioned in **Related Works** “Transformers for decision making” are worthy of applying our methods, which are beyond the goal of this work.
>
> ***Q2**: The baselines used for comparison seem a bit outdated. There have been much stronger baselines in offline RL in recent years, especially diffusion-based methods such as Diffusion-QL and DD. The authors should include these as stronger baselines.*
>
> **A2**: Thank you very much for your constructive opinion. In recent work focused on Decision Transformer **[1][2]**, we find them not using diffusion policy as the baseline, which may be due to the large gap compared with DT. However, as recent strong baselines, we think they are worth comparing. We have tried it out on our benchmarks, shown in **Appendix F, Table 12**, prominently showing the power of our method under poor-data regime.
>
> ***Q3**: The experiments only include the medium datasets in D4RL and are missing the medium-replay and medium-expert datasets. I expect LaMO to perform well on medium-replay as these datasets are of low-quality and should highlight the advantage of language pretraining.*
>
> **A3**: We have conducted experiments on Medium-Replay and Medium-Expert datasets, shown in **Appendix F, Table 8,9**, the results solidate the advantages of our method. As mentioned in **Appendix B**, we explored both Complete and Partial datasets in the Kitchen environment, while we focused on Medium datasets in the MuJoCo and Atari environments. We emphasize again: Medium data sets are mainly used in our work because they meet two properties that we want: 1. low-quality 2. fair and robust for sampling in poor data regime, as we stated in **Appendix B**, "The utilization of this dataset is aimed at minimizing variations in quality among different trajectories".
>
> ***Q4**: The first paragraph of Section 4.1 is a bit confusing, it made me think that the authors pretrained GPT-2 themselves. It should be stated clearly that the authors used the pretrained GPT-2 from HF and only pretrained its special variants.*
>
> **A4**: Thank you for pointing out that our statements are a bit confusing, and we have rephrased it in our updated paper.
>
> ***Questions***:
>
> ***Q5**: Can we replace the language loss with a regularization loss so that the fine-tuned model is not too far away from the original pre-trained model? For example, we can minimize the discrepancy between the embeddings of the fine-tuned and pre-trained models.*
>
> **A5**: In this work we actually use language loss as a regularization term, which is particularly effective in certain tasks, as we stated in Section 5.5. Minimizing embedding discrepancy is interesting and reasonable, but we think it cannot be used in our model in a straightforward way: We are using two distinct sets of embedding for language and motion control respectively. As a matter of the fact, the method of reducing the discrepancy of two embeddings has been tried by Wiki-RL, where they utilize K-means clustering and similarity loss, which turns out not to be very helpful, as stated in their paper **[3] Section 5.6 Table 6**.
>
> [1] Wu, et al. “Elastic Decision Transformer” arXiv preprint arXiv:2307.02484
>
> [2] Yamagata, et al. “Q-learning Decision Transformer: Leveraging Dynamic Programming for Conditional Sequence Modeling in Offline RL” arXiv preprint arXiv:2209.03993
>
> [3] Reid, et al. “Can Wikipedia Help Offline Reinforcement Learning?” arXiv preprint arXiv:2201.12122

---

> > ### Comment · Reviewer_1Dwk · 2023-11-22
> >
> > I thank the authors for the response and the additional experiments, which have addressed some of my concerns. However, my main concern regarding the applicability of the framework to other LLMs and RL models still remains. Therefore, I'll keep my original score, which leans toward acceptance.
> >
> > Question:
> > - In Table 8 and 9 why did you test LaMO on two different ratios?

---

> > > ### Author Response · Authors · 2023-11-23
> > > **Thank you for the feedback!**
> > >
> > > We are happy that our replies address some of your concerns! We would like to answer your question as follows:
> > >
> > > *In Table 8 and 9 why did you test LaMo on two different ratios?*
> > >
> > > To emphasize the advantage of LaMo in data-poor scenarios, we choose the data ratio to be 1% for the Medium-Replay dataset, and the results show the consistent advantages of LaMo with low-quality data. As for the Medium-Expert dataset, the mixing of expert demonstrations and suboptimal data leads to substantial variance during downsampling, and thus we use the original dataset.
> > >
> > > *However, my main concern regarding the applicability of the framework to other LLMs and RL models still remains.*
> > >
> > > Our framework (sequentially pre-training+deeper embedding+LoRA+language loss) is adaptable to transformer-based RL methods such as **[1][2]**. We have explored applying LLaMA to our framework, and we anticipate that our work could motivate the community to delve into further applications of stronger LMs in offline RL.
> > >
> > > Again, we sincerely thank you for the constructive suggestions.
> > >
> > > [1] Janner, et al. “Offline Reinforcement Learning as One Big Sequence Modeling Problem” arXiv preprint arXiv:2106.02039
> > >
> > > [2] Liu & Abbeel. “Emergent Agentic Transformer from Chain of Hindsight Experience” arXiv preprint arXiv:2305.16554

---

### Official Review · Reviewer_mYMk · 2023-11-01

**Soundness:** 3 good
**Presentation:** 4 excellent
**Contribution:** 2 fair
**Rating:** 5
**Confidence:** 4

**Summary:**

This paper studies the use of decision transformer (DT) and pretrained language models for offline RL control. The authors propose a new framework Language Models for Motion Control (LaMo) that improves upon a naive use of a pretrained LM for DT training. The framework 1) uses a pretrained LM model, 2) uses LoRA fine-tuning, 3) use higher capacity input embedding and output networks and 4) use auxiliary language prediction loss during finetune. Empirical results are provided to show the proposed method outperform other transformer-based methods and also offline RL methods in a number of benchmark tasks.

**Strengths:**

**originality**
- Main novelty of the paper is an improved framework to finetune pretrained LM on RL task. Compared to the Reid et al paper, the main additions are the finetuning technique and the increased capacity of the projection layer for input embeddings and output layer. Though the changes are relatively simple, they are shown to provide much stronger performance, can be a novel contribution.

**quality**
- technical details provided for reproducing the results.
- good discussion of related works

**clarity**
- paper is written clearly and easy to follow

**significance**
- some insights are provided on potential reason that DT methods work better on sparse reward setting.
- ablations showing the contribution of each component.
- the proposed changes are not too complex, I appreciate the simplicity.
- empirical resutls show significant improvement over DT and DT+Wiki over all 3 benchmarks.

**Weaknesses:**

Comparisons:
- Figure 1, in Reid et al paper, they show that a pretrained DT tend to give improved performance, why in your figure DT gets better performance in Kitchen and Atari?
- Did you finetune the methods you are comparing to on the benchmarks you are studying?
- The other thing is only some of the tasks in each benchmark are studied, so it is also a bit concerning whether there will still be a big perforrmance gap between proposed method and baseline when other tasks are also tested.

Novelty and significance:
- Technical novelty of the method is a bit lacking. Essentially compared to DT+Wiki, a different finetuning method is used, and the projection and output layers are made bigger. Neither of these are new techniques.

**Questions:**

It is unclear why language pretraining can help RL tasks which have a large domain gap?

---

> ### Author Response · Authors · 2023-11-19
> **Response to Reviewer mYMk --- Part 1/2**
>
> We thank the reviewer for the constructive comments and suggestions. We address each of your comments in the following.
>
> ***Weaknesses***:
>
> ***Q1**: Figure 1, in Reid et al paper, they show that a pre-trained DT tends to give improved performance, why in your figure DT gets better performance in Kitchen and Atari?*
>
> **A1**: We strongly agree with the point that “pre-trained DT tends to give improved performance”, while we would like to add that techniques proposed in our work are actually required to “unleash” the power of pre-trained DT. In MuJoCo tasks, Wiki-RL gains higher scores in each task compared with DT, which is both examined by **[1]** and our work. However, their work has not tested Wiki-RL in the Kitchen environment, which is harder than MuJoCo. In the Atari environment, the dataset they adopt (1% DQN-replay dataset) is different from ours (100%, 10% D4RL Medium dataset). After running extensive experiments, we find that Wiki-RL could not prominently leverage the power of pre-trained DT in these two difficult settings although they obtain advantages in MuJoCo tasks. We propose this is due to the fact that fully fine-tuning the pre-trained model could compromise the general knowledge of pre-trained models, and we find that full fine-tuning could not work at all in certain tasks. This phenomenon is presented in the ablation part **Appendix G, Fig 13**.
>
> ***Q2**: Did you finetune the methods you are comparing to on the benchmarks you are studying?*
>
> **A2**: Yes, we have looked into the hyperparameters configuration of baselines on our benchmarks, which investigate the poor data regime of D4RL. Specifically, we run the baseline on MuJoCo and Kitchen environments based on CORL (https://github.com/tinkoff-ai/CORL) and run the baseline on Atari using d3rlpy (https://github.com/takuseno/d3rlpy). For both DT-based and value-based methods, we tuned parameters on representative tasks from each domain of our benchmark. It is crucial to note that, within the range of settings of hyperparameters explored, the current parameters adopted by our baselines have been identified as not worse than others, and the results are shown in **Appendix F, Fig 11**.
>
> ***Q3**: The other thing is only some of the tasks in each benchmark are studied, so it is also a bit concerning whether there will still be a big performance gap between proposed method and baseline when other tasks are also tested.*
>
> **A3**:  To ensure strong alignment with the existing literature on DT and offline RL research, we selected tasks identical to those used in **[1][2][3][4]**, including MuJoCo domains such as Hopper, Walker2d, Halfcheetah, Reacher, and Atari domains like Breakout, Qbert, and Pong. Additionally, we included a challenging domain, Kitchen, where our approach demonstrated promising results. It is crucial to emphasize that our choice of tasks is not due to any limitations in applicability to other tasks; rather, we specifically focused on tasks that are most commonly encountered in this research direction, ensuring maximum comparability. To solidify this point, we conduct additional tests on tasks Freeway, Asterix in Atari and Ant in MuJoCo, as shown in **Appendix F, Table 10,11**, which further prove the claims made in our **Experiments** part.
>
> ***Q4**: Technical novelty of the method is a bit lacking. Essentially compared to DT+Wiki, a different finetuning method is used, and the projection and output layers are made bigger. Neither of these are new techniques.*
>
> **A4**:
>
> We acknowledge that the implementations we utilized, such as LoRA, MLP embedding, and language loss, are neither novel nor overly complex. However, we would like to underscore that the novelty of our work does not solely lie in any individual module we employed. Instead, our innovation stems from the exploration of the scientific question—whether language pre-training can help offline RL, which is inherently novel and insightful. We successfully demonstrated the potential of this direction by effectively combining simple existing methods, highlighting this as the distinctive aspect of our work.
>
> As articulated in **Related Works** “LLM for decision making” part, many recent studies focus on using language models as high-level planners or are limited to tasks primarily involving textual input. In contrast, our approach investigates the possibility of efficiently leveraging the inductive bias of language models for low-level motion control without expanding model parameters. We provide an affirmative answer to this question. Furthermore, devising and correctly combining these implementations is non-trivial, as evidenced by our ablation experiments. Incorrectly using the pre-trained model resulted in suboptimal performance. Our framework, which combines all these implementations and adeptly adapts the pre-trained model, unleashes the potential of pre-trained language models for offline RL.

---

> > ### Author Response · Authors · 2023-11-19
> > **Response to Reviewer mYMk --- Part 2/2**
> >
> > ***Questions***:
> >
> > ***Q5**: It is unclear why language pre-training can help RL tasks which have a large domain gap.*
> >
> > **A5**: Your question is indeed insightful. Exploring this point is the main goal and highlight of our work. In **Section 5.5** “Language pre-training v.s. visual pre-training”, we attribute it to “**proficiency in sequential modeling is the key to unleashing the potential of cross-domain pre-trained models**”. We have conducted extensive experiments, replacing the pre-trained model with image pre-trained, early-stopped pre-trained, random corpus pre-trained, and random weights initialized. Empirically, all of those results indicate that: two properties, 1. high-quality pre-trained and 2. pre-trained on sequential modeling tasks, are actually required to prominently unleash the power of pre-trained models in our method. As recent works **[5][6]** proposed,  language model could be experts in motion control if we view motion control as a downstream task in sequential modeling, the same as language modeling, and we emphasize again that, our method can effectively use the prior knowledge acquired in language pre-training to deal with offline RL.
> >
> > [1] Reid, et al. “Can Wikipedia Help Offline Reinforcement Learning?” arXiv preprint arXiv:2201.12122
> >
> > [2] Chen, et al. “Decision Transformer: Reinforcement Learning via Sequence Modeling” arXiv preprint arXiv:2106.01345
> >
> > [3] Kumar, et al. “Conservative Q-Learning for Offline Reinforcement Learning” arXiv preprint arXiv:2006.04779
> >
> > [4] Kostrikov, et al. “Offline Reinforcement Learning with Implicit Q-Learning” arXiv preprint arXiv:2110.06169
> >
> > [5] Brohan, et al. “RT-2: Vision-Language-Action Models Transfer Web Knowledge to Robotic Control” arXiv preprint arXiv:2307.15818
> >
> > [6] Wang, et al. “Prompt a Robot to Walk with Large Language Models” arXiv preprint arXiv:2309.09969

---

> > > ### Author Response · Authors · 2023-11-22
> > > **Thank you for the review and awaiting your response**
> > >
> > > As the end of the author-reviewer discussion is approaching, we kindly ask for your review of our revised paper and response. If our revisions have adequately addressed your concerns, we'd greatly appreciate your consideration in adjusting the scores. Should you have any remaining questions or require further clarification, please don't hesitate to let us know. We would be more than happy to answer any further questions.

---

### Official Review · Reviewer_GHKm · 2023-11-01

**Soundness:** 3 good
**Presentation:** 4 excellent
**Contribution:** 3 good
**Rating:** 8
**Confidence:** 4

**Summary:**

The authors observe shortcomings in previous applications of pre-trained (transformer) language models (LM) in reinforcement learning (RL) and in particular in control: they are only used as initializations or as interfaces and do not outperform non-pre-trained transformer models such as decision transformers (DT).

They propose LoMa which adds 3 components to better leverage (or "unleash") pre-trained LMs in RL: 1) replacing the linear embedding with trainable MLPs, 2) performing low-rank adaptation (LoRA) for the RL training, and 3) maintaining the original language model loss while training with RL.

The authors claim that LoMa outperforms competitive offline RL baselines and DT on sparse-reward tasks and in low-data regimes.

**Strengths:**

The paper attempts to overcome important shortcomings of leveraging foundation models (here LMs) in RL in particular offline RL.

Originality:
- Adding the sparse-reward experiments compared to the dense-reward environments in the Wiki-RL reference paper [1] to show the benefits of using pre-trained models is original and well-motivated.
- Using LoRA to fine-tune the LM to a policy is interesting, ~~although not well-motivated.~~
- The ablation with a model pre-trained on a random corpus to show the importance of sequential modeling is original.


Clarity:
- The motivation and the contributions of the paper are very clear.
- The structure of the paper is easy to follow.
- Overall the paper is well written.

Significance:
- LaMo performs competitively in the low-data regime as opposed to Wiki-RL showing that it potentially leverages/”unleashes” pre-trained LMs in a better way.
- Overfitting, and other capacity loss phenomena, arising when more gradient steps are performed on the same data are major problems in offline RL. Observing that the auxiliary language loss (and maybe LoRA) introduced in LaMo helps in overcoming this issue is relevant beyond the scope of this paper.

[1] Reid, Machel, Yutaro Yamada, and Shixiang Shane Gu. "Can Wikipedia help offline reinforcement learning?." arXiv preprint arXiv:2201.12122 (2022).

**Weaknesses:**

*(update: all the weaknesses below have been addressed by the authors.)*

The points about the experimental results raised below and the questions raised in the question sections make it hard for me to confirm the validity of the claim made in the paper that LoMa outperforms baselines and that its components are crucial.
Clarifications on the experimental setup and results if valid would help raise my soundness and overall score.

**Major:**

1)
I have a major concern over the validity of the experimental protocol and results presented in the paper from the observation that the performance reported for the baselines differs significantly from their performance as reported in their original papers. In particular
- BC and CQL in the D4RL paper [2] obtain scores significantly divergent from the scores reported in this paper. Can the authors explain this divergence, and if due to different training budgets/settings explain the motivation behind the different settings?
E.g. CQL and BC achieve 43.8 and 33.8, respectively, on Franka-complete in D4RL [2], but are reported to achieve 0 scores in this paper. (Overfitting of CQL is mentioned, but does not clarify the discrepancy)
- Also BC and CQL are reported to perform better on Kitchen with 1% of the datasets (or other fractions) than on Kitchen with full datasets. Same for Wiki-RL and DT on some specific splits.
- Similarly, Wiki-RL is reported to perform worse than the same model (ChibiT) in its original paper on Atari. Whereas DT is reported to perform better than in the wiki-RL paper. E.g. in Breakout 350 DT  and 130 Wiki-RL in this paper vs 267 DT and 280 Wiki-RL in the Wiki-RL paper.

2)
 The ablation studies are presented each time on a different task&dataset-ratio combination and the full ablation results are not in the appendix (only the pre-training ablation is given). This can lead to a biased evaluation of LaMo if only good results are presented.
The combination of components of LaMo is a major contribution of the paper, claimed to "unleash" the potential of pre-trained LMs, and ablation experiments are thus crucial to assess the validity of this claim and the significance of the paper.


**Significant:**
- I don't find LoRA to be well-motivated, or at least that its benefits are. The authors mention early overfitting but this is not clear from Figure 5 (LoMa also experiences small drops in performance).
-  Several elements of the paper can be mistakenly interpreted as original contributions of the paper, such as the language modeling loss (also used in Wiki-RL), the ablation on ImageGPT (also used in Wiki-RL), the selection of the baseline (similar to Wiki-RL).
While not being novel was not the issue, it was often not clear if those contributions were original.
- It’s hard to assess the significance of an improvement when curves are shown with shaded areas in [μ − 0.5σ, μ + 0.5σ].

**Minor:**
- Notation is not ambiguous for readers familiar with the topics but would benefit from better presentation. Some variables are undefined in some equations e.g. the expectation distributions are underspecified in equation 1, $a'$ in equation 3, $T$ in equation 4.
- Citations don’t include the journal/proceedings details which makes it hard to identify which version of a paper was used in some cases.

[2] Fu, Justin, et al. "D4rl: Datasets for deep data-driven reinforcement learning." arXiv preprint arXiv:2004.07219 (2020).

**Questions:**

*(update: all the questions below have been addressed by the authors.)*

**Major:**

- Can the authors point to the specifications of the Reacher2d environment they used? The Reacher I’m familiar with[2] would not be considered a sparse-reward environment. This would also help to confirm the expert score.
It would also help to have the score of the policy that generated the medium dataset.
- Can the authors indicate which Atari offline datasets they have used? D4RL does not seem to provide Atari datasets in its original version.
- How much hyperparameter tuning has been spent on the value of the language loss hyperparameter and the fraction of parameters to train with LoRA?
- Are the hyperparameters in Appendix E, used with all transformers in the paper? (LoMa, Wiki-RL, DT)?
- Do the transformers and non-transformers used in the paper have a comparable number of parameters to ensure a fair comparison of performance?

**Minor:**
- Figure 6(a): why do the curves not start from the same point at training step 0? How can the authors explain that the cross-entropy loss decreases significantly for an already pre-trained model (red curve)? and also eventually decreases for the ablation (blue curve)

[3] https://gymnasium.farama.org/environments/mujoco/reacher/

---

> ### Author Response · Authors · 2023-11-19
> **Response to Reviewer GHKm --- Part 1/4**
>
> We thank the reviewer for the constructive comments and suggestions. We address each of your comments in the following.
>
> ***Weaknesses***:
>
> ***Major***:
>
> ***Q1**: I have a major concern over the validity of the experimental protocol and results presented in the paper from the observation that the performance reported for the baselines differs significantly from their performance as reported in their original papers. In particular，*
>
> ***Q1.1**: BC and CQL in the D4RL paper obtain scores significantly divergent from the scores reported in this paper. Can the authors explain this divergence, and if due to different training budgets/settings explain the motivation behind the different settings? E.g. CQL and BC achieve 43.8 and 33.8, respectively, on Franka-complete in D4RL, but are reported to achieve 0 scores in this paper. (Overfitting of CQL is mentioned, but does not clarify the discrepancy)*
>
> **A1.1**:
>
> Yes, the difference arises from the reported metric. Some offline RL papers use **the performance of the best several checkpoints**, while for more scientific study, we use “**the average performance over the last 20K training steps out of a total of 100K training steps with evaluations conducted every 2500 training steps**” as stated in **Section 5.1**. This is because in the context of offline RL, the RL agent could never interact with the environment during training， and thus mitigating overfitting is a crucial aspect of a method's effectiveness. This choice aligns with our perspective on the importance of robustness in the evaluation of offline RL methods.
>
> Therefore, when an algorithm exhibits overfitting on a certain task (shown in **Appendix F, Fig 9**) after an excessive number of gradient steps, differences in metrics can result in discrepancies between our results and the values reported in the D4RL paper. We are also willing to report the results on the metric of top performance, as shown in **Appendix F, Table 13**, which shows the consistent advantages of our method and solidates the validity of our claims.
>
> ***Q1.2**: Also BC and CQL are reported to perform better on Kitchen with 1% of the datasets (or other fractions) than on Kitchen with full datasets. Same for Wiki-RL and DT on some specific splits.*
>
> **A1.2**:
>
> Yes, this is reasonable because of the property of the Kitchen dataset: the Partial dataset contains data that cannot complete the task and data that can, as described in **Appendix B**. Therefore, when the dataset is being downsampled, the number of varieties of data would decrease and thus could be easier for the model to learn. This explains why algorithms could perform better with less data in certain cases.
>
> Besides, as mentioned in **A1.1**, there also exists the issue of overfitting. In cases of overfitting, comparing the weak performance lacks statistical meaning.
>
> ***Q1.3**: Similarly, Wiki-RL is reported to perform worse than the same model (ChibiT) in its original paper on Atari. Whereas DT is reported to perform better than in the wiki-RL paper. E.g. in Breakout 350 DT and 130 Wiki-RL in this paper vs 267 DT and 280 Wiki-RL in the Wiki-RL paper.*
>
> **A1.3**: The scores reported in Wiki-RL are different from those in our paper because we use different datasets. Wiki-RL runs its experiments on the 1% DQN-replay Atari dataset, while we run our experiments on the D4RL-Atari medium dataset, of different data ratios.
>
> ***Q2**: The ablation studies are presented each time on a different task & dataset-ratio combination and the full ablation results are not in the appendix (only the pre-training ablation is given). This can lead to a biased evaluation of LaMo if only good results are presented. The combination of components of LaMo is a major contribution of the paper, claimed to "unleash" the potential of pre-trained LMs, and ablation experiments are thus crucial to assess the validity of this claim and the significance of the paper.*
>
> **A2**: We enhance the ablations during the rebuttal phase and we hope that these full and consistent ablation experiments could show the consistent advantages of our proposed techniques in leveraging the pre-trained language model for offline RL tasks, as shown in **Appendix G**.

---

> > ### Author Response · Authors · 2023-11-19
> > **Response to Reviewer GHKm --- Part 2/4**
> >
> > ***Significant***:
> >
> > ***Q3**: I don't find LoRA to be well-motivated, or at least that its benefits are. The authors mention early overfitting but this is not clear from Figure 5 (LoMa also experiences small drops in performance).*
> >
> > **A3**:
> >
> > For the motivations,  1. As we view motion control as a downstream task in sequential modeling, LoRA, as a well-known parameter-efficient fine-tuning technique for adapting downstream tasks, is expected to leverage the pre-trained model well.  2. There are works **[1]** showing that fine-tuning only a small portion of parameters could outperform frozen and full fine-tuning.
> >
> > We have provided more ablation results on LoRA compared with full finetuning and freezing, as shown in **Appendix G, Fig 13**.  We observe that in our settings, freezing the pre-trained parameters and adapting with LoRA could not only improve training efficiency but also address the issue of overfitting that occurs in full finetuning. **Notably, our experiments show that full finetuning could not work at all in certain tasks**, and freezing all parameters in Transformer could restrict the learning ability of the mode.
> >
> > ***Q4**: Several elements of the paper can be mistakenly interpreted as original contributions of the paper, such as the language modeling loss (also used in Wiki-RL), the ablation on ImageGPT (also used in Wiki-RL), the selection of the baseline (similar to Wiki-RL). While not being novel was not the issue, it was often not clear if those contributions were original.*
> >
> > **A4**:
> >
> > The language modeling loss term was proposed in **[2]**,  however, its ablation experiments in **Section 5.6 Table 6** of that paper have not prominently shown the improvement brought by this term. Built on their work, we further demonstrate its role in preventing overfitting in certain tasks by ablation study, and show that the pre-trained language model could master the knowledge of two domains: language and motion at the same time. To prevent being mistakenly interpreted, we have added a citation of Wiki-RL in the method part for this implementation.
> >
> > As for the ablation on imageGPT and the choice of baselines, we did not regard them as the novelty of our work. We actually use different baselines from Wiki-RL, like IQL and BCQ, which are the most commonly used powerful baselines in offline RL, and notably, we also compare with Wiki-RL itself, as a baseline utilizing pre-trained models.
> >
> > ***Q5**: It’s hard to assess the significance of an improvement when curves are shown with shaded areas in [μ − 0.5σ, μ + 0.5σ].*
> >
> > **A5**: Thank you for your suggestion. We have provided more ablation results with shaded areas in [μ − σ, μ + σ] in **Appendix G**.
> >
> > ***Minor***:
> >
> > ***Q6**: Notation is not ambiguous for readers familiar with the topics but would benefit from better presentation. Some variables are undefined in some equations e.g. the expectation distributions are underspecified in equation 1, a′ in equation 3, T in equation 4.*
> >
> > **A6**: Thank you for your advice, we have fixed these undefined notations in equation 1,3,4 in our updated paper.
> >
> > ***Q7**: Citations don’t include the journal/proceedings details which makes it hard to identify which version of a paper was used in some cases.*
> >
> > **A7**: We have revised the format of the reference to reflect the journal/proceedings details in our updated paper.

---

> > > ### Author Response · Authors · 2023-11-19
> > > **Response to Reviewer GHKm --- Part 3/4**
> > >
> > > ***Questions***:
> > >
> > > ***Major***:
> > >
> > > ***Q8**: Can the authors point to the specifications of the Reacher2d environment they used? The Reacher I’m familiar with would not be considered a sparse-reward environment. This would also help to confirm the expert score. It would also help to have the score of the policy that generated the medium dataset.*
> > >
> > > **A8**:
> > >
> > > We use the Reacher2d environment directly from the code repository of Decision Transformer (https://github.com/kzl/decision-transformer). It is clarified in that **page 7 of [3]**, “Reacher is a goal-conditioned task and has sparse rewards, so it represents a different setting than the standard locomotion environments (HalfCheetah, Hopper, and Walker). ”
> > >
> > > The dataset for Reacher2d is designed by ourselves, which is described in detail in Appendix B, and the expert score is directly set as 100 for simplicity, described in **Appendix D**.
> > >
> > > The score of the policy that generated the medium dataset is 36.0. Thank you for your suggestions and we’ve added this detail into **Appendix B**.
> > >
> > > ***Q9**: Can the authors indicate which Atari offline datasets they have used? D4RL does not seem to provide Atari datasets in its original version.*
> > >
> > > **A9**: We use d4rl-atari (https://github.com/takuseno/d4rl-atari). We have updated it in **Section 5.1** and **Appendix B**.
> > >
> > > ***Q10**: How much hyperparameter tuning has been spent on the value of the language loss hyperparameter and the fraction of parameters to train with LoRA?*
> > >
> > > **A10**: We did not invest extensive effort in selecting these two hyperparameters. For the coefficient lambda in the language loss, we only experimented with values of 0.1 and 1. To determine the hyperparameter lora-dim that governs the parameter scale for LoRA, we explored values of 16, 32, and 64, ultimately choosing 32 for Atari, and 16 for MuJoCo, Kitchen.
> > >
> > > ***Q11**: Are the hyperparameters in Appendix E, used with all transformers in the paper? (LaMo, Wiki-RL, DT)?*
> > >
> > > **A11**:
> > >
> > > Yes, the context length, return-to-go, training steps, batch size, dropout are shared by all these DT-based methods. And our method utilizes the approximately same model size as Wiki-RL, including the same hidden size, number of layers, except for the MLP embedding part stated in Method.
> > >
> > > Besides, the learning rate and weight decay in Appendix E are not directly used in Wiki-RL and DT as our approach adopts a pre-train + parameter-efficient fine-tune approach, while DT and Wiki-RL both adopt full supervised training. We have tuned DT across different tasks to obtain good convergence performance among explored configurations, as shown in **Appendix F, Fig 11**. As for Wiki-RL, we sincerely follow the authors’ statements in their paper **[2] Section 4.1**, to use the same learning hyperparameters as DT.
> > >
> > > ***Q12**: Do the transformers and non-transformers used in the paper have a comparable number of parameters to ensure a fair comparison of performance?*
> > >
> > > **A12**:
> > >
> > > The numbers of parameters of DT, Wiki-RL, and our method are 7.3M, 125M, and 128M, respectively. And for non-transformers (value-based) methods, the numbers are relatively small, such as 0.41M for CQL.
> > >
> > > For value-based methods, when the model parameters are relatively large, training becomes extremely challenging, leading to results worse than smaller models. Therefore, for value-based baselines, we adhere to widely accepted model sizes, which are much smaller than Transformer-based methods.
> > >
> > > It is important to emphasize that simply increasing the size of the Transformer will not boost the performance, as shown in the results of Wiki-RL and our ablation studies. Moreover, although our method involves a relatively large model, the number of trainable parameters is fairly small, which is 3.5M. So the difference of the number of parameters between transformers and value-based methods does not compromise the fairness of the comparisons of performance.

---

> > > > ### Author Response · Authors · 2023-11-19
> > > > **Response to Reviewer GHKm --- Part 4/4**
> > > >
> > > > ***Minor***:
> > > >
> > > > ***Q13**: Figure 6(a): why do the curves not start from the same point at training step 0? How can the authors explain that the cross-entropy loss decreases significantly for an already pre-trained model (red curve)? and also eventually decreases for the ablation (blue curve)*
> > > >
> > > >  **A13**:
> > > >
> > > > The curve does not start from the same point because we did not immediately evaluate after initialization. Actually, the starting point reflects the results after training for 2500 training steps. To enhance clarity, we have added a point at the beginning of the curve.
> > > >
> > > > The continued decrease in loss is attributed to the jointly trained model utilizing a small text dataset, WikiText, which introduces a distribution shift compared to the pretraining dataset of GPT-2, which is a very large corpus. This shift allows the model to further adapt, thus the entropy loss could decrease.
> > > >
> > > > It is important to note that the phenomenon of loss eventually decreasing in the ablation is not consistent. We provide an additional example that demonstrates a different result, as shown in **Appendix G, Fig 14(a)**.
> > > >
> > > > [1] Ze, et al. "H-InDex: Visual Reinforcement Learning with Hand-Informed Representations for Dexterous Manipulation." arXiv preprint arXiv:2310.01404
> > > >
> > > > [2] Reid, et al. “Can Wikipedia Help Offline Reinforcement Learning?” arXiv preprint arXiv:2201.12122
> > > >
> > > > [3] Chen, et al. “Decision Transformer: Reinforcement Learning via Sequence Modeling” arXiv preprint arXiv:2106.01345

---

> ### Comment · Reviewer_GHKm · 2023-11-21
> **Most concerns resolved. One remaining.**
>
> I thank the authors for their complete answers to all my concerns. Most of them have been resolved, improving the presentation of the paper and increasing my confidence that the experiments support the claims made in the paper (although I cannot validate the numbers themselves without running the code. In the future, I would strongly encourage the authors to share their code or maybe share tracking results from an experiment tracking service like Weights & Biases).
>
> I have one concern remaining regarding the size of the models compared which I'd like the authors to address.
>
> In their reply, the authors say that
> (**A12:**) *"The numbers of parameters of DT, Wiki-RL, and our method are 7.3M, 125M, and 128M, respectively. And for non-transformers (value-based) methods, the numbers are relatively small, such as 0.41M for CQL."*
> Revealing a large gap in the number of parameters of the models compared, even within the same family of transformers DT vs LaMo.
> However, the authors also point out that the comparison should not be made on the basis of the total number of parameters but on the basis of the number of trainable parameters, which in that case would be 3.5M for LaMo and comparable to DT.
>
> I appreciate this distinction, but believe that it could benefit from an explicit presentation in the paper, so that future readers do not extrapolate or misunderstand from the paper that using LaMo would result in better performance than training a DT model of the same size.
> It would also be beneficial to state the number of (trainable) parameters of all models and baselines in the appendix.
>
> Regarding value-based models I believe that the following claim would need a reference:
> *"For value-based methods, when the model parameters are relatively large, training becomes extremely challenging, leading to results worse than smaller models. Therefore, for value-based baselines, we adhere to widely accepted model sizes, which are much smaller than Transformer-based methods."*
>
> With both of these resolved (making it clear that it's trainable sizes that are compared, listing them, and justifying the size of the value-based ones), I would be happy to significantly increase my score.
>
> *Minor concern*.
> *A10: "To determine the hyperparameter lora-dim that governs the parameter scale for LoRA, we explored  values of 16, 32, and 64, ultimately choosing 32 for Atari, and 16 for MuJoCo, Kitchen."*
> Can this be reflected in table 7?

---

> > ### Author Response · Authors · 2023-11-22
> > **Thank you for the feedback!**
> >
> > We are happy that our replies address most of your questions! We appreciate your feedback and have updated our paper in response to your concerns:
> >
> > 1. Clarifying and listing the number of trainable sizes of our method and baselines in **Appendix F, Table 14**. We agree that comparisons should be made on the basis of trainable parameters, and have indicated this point when comparing LaMo with DT in **the last paragraph of Appendix F**. We have also provided the total number of parameters of each method for reviewers’ and readers’ reference in this table.
> >
> > 2. Revising the statements about the model sizes of value-based ones in our paper, and providing reference [1] and experiments results in **Appendix F, Figure 16** (*To avoid affecting the numbering of figures in the contents of the rebuttal, we have temporarily put it in the end of sections*) for selections of sizes of value-based methods in **the last paragraph of Appendix F**. Specifically, **Section 3 and the ablation study in Section 4.5 in [1]** demonstrate that deepening the network structure does not yield improvements for the value-based method, and our experiments reveal a similar trend for increasing the network's width.
> >
> > 3. Reflecting the choice of LoRA rank we used in **Appendix E, Table 7**.
> >
> > Besides, **as we stated in Appendix A, we are committed to releasing the code.** Again, we sincerely thank you for the constructive suggestions.
> >
> > [1] Tarasov, et al. “Revisiting the Minimalist Approach to Offline Reinforcement Learning” arXiv preprint arXiv:2305.09836

---

> > > ### Comment · Reviewer_GHKm · 2023-11-22
> > > **All concerns resolved. Raising my score from 5 to 8.**
> > >
> > > I thank the authors for addressing and resolving all the concerns I had. I have raised my score from 5 to 8 and updated my review.

---

> > > > ### Author Response · Authors · 2023-11-22
> > > > **We appreciate your quick and positive feedback!**
> > > >
> > > > We're delighted to know that our responses have addressed your concerns. We deeply appreciate the reviewer's kind response and positive feedback.

---

### Author Response · Authors · 2023-11-19
**General Response to All Reviewers**

We thank all the reviewers for their insightful comments. We have addressed all your individual comments. We want to thank the reviewers for acknowledging the novelty and empirical evaluation of our work - “*performs competitively in the low-data regime*” (GHKm), “*much stronger performance, can be a novel contribution*” (mYMk), “*the idea is intuitive and simple*” (1Dwk), “*the results span a diverse array of scenarios*” (ei7j). More additional experiments are also conducted during the rebuttal phase to support our proposed method (given in the updated PDF file), as suggested by the reviewers. The updates are marked in blue.

**EXP1: Complete Experimental Results of the Ablation Study** in reply to GHKm and ei7j. Results are shown in **Appendix G**. We conduct an extensive ablation study of each technical component, including MLP embedding, LoRA, language loss, language pre-training, and the quality of pre-trained models as well as corpus. These results present the consistent effectiveness of each component we propose in our methodology.

**EXP2: Training on datasets with varying qualities** in reply to 1Dwk. **Appendix F, Tables 8 and 9** present normalized scores when training on Medium-Expert and Medium-Replay datasets on MuJoCo. LaMo still shows competitive performance over the baselines, especially on Medium-Replay (1%) datasets.

**EXP3: More tasks** in reply to mYMk. Results are provided in **Appendix F, Tables 10 and 11**. In Atari games Freeway and Asterix, LaMo effectively narrows the performance gap between DT-based methods and value-based methods, while in MuJoCo Ant environment, LaMo reaches the best performance, surpassing the baselines.

**EXP4: Comparing with Diffusion-QL** in reply to 1Dwk. Results are shown in **Appendix F, Table 12**. A comparative analysis is presented between our method and the recent powerful diffusion policy, Diffusion Q-learning, across three tasks. Notably, our method outperforms Diffusion-QL in the low-data regime.

**EXP5: Overfitting and top-k metric** in reply to GHKm. We show additional results indicating that CQL encounters the problem of overfitting in **Appendix F, Fig 9**. We also provide the experiment results of Kitchen based on the top-k metric in **Appendix F, Table 13**, which calculates the average scores over the k checkpoints with the highest testing scores. LaMo still outperforms other DT-based methods and value-based methods.

**EXP6: Effects of model size** in reply to ei7j. As is shown in **Appendix F, Fig 10**, the influence of language model size on the performance of LaMo is depicted. The results indicate that GPT2-small already achieves satisfactory performance. Additionally, in specific tasks, GPT2-medium demonstrates a slight edge over GPT2-small, showing incremental improvements with increased model size.

**EXP7: Hyperparameter tuning for baselines** in reply to to GHKm and mYMk. In **Appendix F, Fig 11**, we conduct hyperparameter tuning for both DT-based and value-based baselines. These results verify the baselines’ performance reported in our study.

Again, we thank the reviewers for their thoughtful feedback and are happy to address any further comments from reviewers.

---

### Meta-Review · Area_Chair_XP2G · 2023-12-12

**Metareview:**

### Summary

The paper "Unleashing the Power of Pre-trained Language Models for Offline Reinforcement Learning" introduces LaMo, a framework utilizing pre-trained Language Models (LMs) for offline RL. It enhances RL performance by incorporating Decision Transformers, employing LoRA fine-tuning, non-linear MLP transformations for embeddings, and an auxiliary language prediction loss. LaMo demonstrates superior performance in sparse-reward tasks and bridges the gap between value-based offline RL methods and decision transformers, especially in scenarios with limited data samples.

### Decision
Overall, the paper is well-written, the reviewers raised some confusion, and the authors did a tremendous job at addressing those concerns during the rebuttal. The results are convincing. The method is interesting and reasonable. This paper would be of interest to the ICLR community. I couldn't identify any unaddressed major concerns raised by the reviewers during the rebuttal period.

**Justification For Why Not Higher Score:**

The reviewers are mostly positive about the paper. The paper wasn't clear initially, but the authors have addressed the issues raised by the reviewers. Some of the reviewers were not very enthusiastic about this paper. Better writing and more ablations would make this paper spotlight-worthy.

**Justification For Why Not Lower Score:**

The paper is interesting and addresses a significant problem. The results are convincing, and it is worth presenting at the ICLR conference.

---

### Decision · Program_Chairs · 2024-01-16

Accept (poster)